# Fast Model Debias with Machine Unlearning

**Ruizhe Chen[1], Jianfei Yang[2], Huimin Xiong[1], Jianhong Bai[1], Tianxiang Hu[1], Jin Hao[3],**
**Yang Feng[4], Joey Tianyi Zhou[5], Jian Wu[1], Zuozhu Liu[1]***

[1] Zhejiang University [2] Nanyang Technological University
[3] Stanford University [4] Angelalign Technology Inc. [5] Centre for Frontier AI Research
ruizhec.21@intl.zju.edu.cn

## Abstract

Recent discoveries have revealed that deep neural networks might behave in a biased manner in many real-world scenarios. For instance, deep networks trained on a large-scale face recognition dataset CelebA tend to predict blonde hair for females and black hair for males. Such biases not only jeopardize the robustness of models but also perpetuate and amplify social biases, which is especially concerning for automated decision-making processes in healthcare, recruitment, etc., as they could exacerbate unfair economic and social inequalities among different groups. Existing debiasing methods suffer from high costs in bias labeling or model re-training, while also exhibiting a deficiency in terms of elucidating the origins of biases within the model. To this respect, we propose a fast model debiasing framework (FMD) which offers an efficient approach to identify, evaluate and remove biases inherent in trained models. The FMD identifies biased attributes through an explicit counterfactual concept and quantifies the influence of data samples with influence functions. Moreover, we design a machine unlearning-based strategy to efficiently and effectively remove the bias in a trained model with a small counterfactual dataset. Experiments on the Colored MNIST, CelebA, and Adult Income datasets along with experiments with large language models demonstrate that our method achieves superior or competing accuracies compared with state-of-the-art methods while attaining significantly fewer biases and requiring much less debiasing cost. Notably, our method requires only a small external dataset and updating a minimal amount of model parameters, without the requirement of access to training data that may be too large or unavailable in practice.

## 1 Introduction

Biased predictions are not uncommon in well-trained deep neural networks [1–3]. Recent findings indicate that many deep neural networks exhibit biased behaviors and fail to generalize to unseen data [4, 5], e.g., convolutional neural networks (CNNs) might favor texture over shape for object classification [6]. For instance, well-trained networks on a large-scale dataset (e.g. CelebA) tend to predict a female person to be with blonde hair, and a male to be with black hair [7, 8]. This is because the number of <blonder hair, female> and <black hair, male> image pairs is significantly higher than others, although there is no causal relationship between hair color and gender [9]. In this case, the model does not learn the correct classification strategy based on human appearance, but rather shows a preference for specific individuals or groups based on irrelevant attributes (error correlations) [2]. Such error correlations not only affect the model's ability to make robust predictions but also perpetuate and exacerbate social bias, resulting in potential risks in many real-world scenarios, such as racism, underestimating minorities, or social disparities among groups in crime prediction [10], loan assessment [11], and recruitment [12] etc.

---

*Corresponding author.

37th Conference on Neural Information Processing Systems (NeurIPS 2023).

Efforts have been made to remove bias in models based on innate or acquired characteristics of individuals or groups. Existing debiasing mechanisms could be categorized into three types depending on when debiasing is conducted: pre-processing, in-processing, and post-processing [2, 13, 14]. Pre-processing debiasing methods usually modify the dataset for fair learning, which often involve reweighing samples [15, 16], modifying feature representations [17, 18], changing labels [19] etc. Another line of research accounts for fairness during training, i.e., in-processing [20–24], including feature-level data augmentation or adversarial training [25, 26] etc. However, the aforementioned methods require expensive costs for human labeling of misleading biases or computationally-intensive debiased model retraining, resulting in unsatisfactory scalability over modern large-scale datasets or models. Few research explore post-processing strategies to achieve fairness with minimal cost [27–29]. They ensure group fairness by alternating predictions of some selected samples, causing degraded accuracy or unfairness on individuals. Moreover, most methods assume that the biased attributes were known, while a generalized debiasing framework should be able to verify whether an attribute (e.g. shape, texture, and color in an image classification task) is biased or not as well [30].

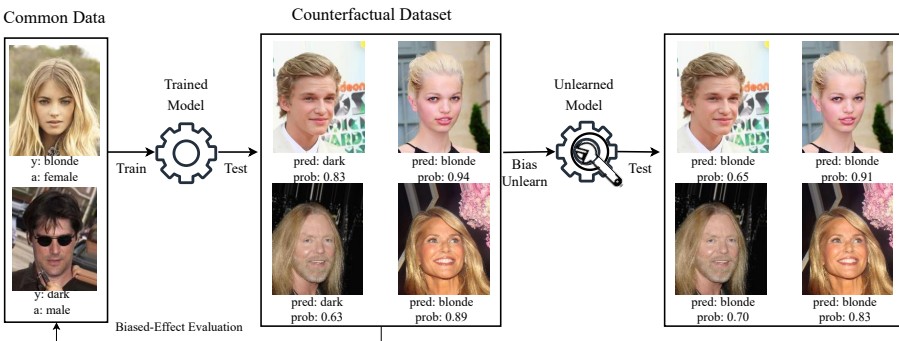

Figure 1: Pipeline of our proposed FMD.

In this paper, we propose FMD, an all-inclusive framework for fast model debiasing. As illustrated in Fig. 1, the FMD comprises three distinct steps: bias identification, biased-effect evaluation, and bias removal. In contrast to pre- or in-processing debiasing methods, our approach eliminates the need for supervised retraining of the entire model or additional labeling of bias attributes. Notably, FMD leverages only a small external dataset, thereby obviating the requirement for access to extensive or unavailable training data in practical scenarios. Furthermore, achieving fair outputs through FMD necessitates updating only a minimal number of parameters, such as the top MLP layers of pre-trained deep networks. Compared to post-processing debiasing methods, FMD yields superior debiasing performance and consistently enhances fairness across diverse bias metrics with little costs.

The FMD operates through the following procedure. Given an attribute and a well-trained model, our first step is to ascertain whether and to what extent the model exhibits bias towards the attribute. To achieve this, we construct a dataset comprising factual samples along with their corresponding counterfactual samples [31], wherein the attribute in question can be varied. By observing how the model's predictions change with the attribute variations, we can effectively identify any bias present. In the biased-effect evaluation phase, we quantitatively assess the extent to which a biased training sample contributes to the model's biased predictions. This evaluation entails measuring how the biased training sample misleads the model and influences its predictions. To this end, we extend the theory of influence functions [32], employing it to estimate the impact of perturbing a biased attribute within the training data on the model's prediction bias measurement. Finally, we introduce an unlearning mechanism that involves performing a Newton step [33] on the learned model parameters to remove the learned biased correlation. We further design an alternative strategy to unlearn biases with the counterfactual external dataset, avoiding hard requirements on access to the training data which might be unavailable in practice. Our unlearning strategy effectively eliminates the estimated influence of the biased attribute, leading to a more fair and unbiased model. Experiments on multiple datasets show that our method can achieve accuracies on par with bias-tailored training methods with a much smaller counterfactually constructed dataset. The corresponding biases and computational costs are significantly reduced as well. Our main contributions are summarized as:

- We propose a counterfactual inference-based framework that can quantitatively measure the biased degree of a trained (black-box) deep network with respect to different data attributes with a novel influence function.

- We propose an unlearning-based debiasing method that effectively and efficiently removes model biases with a small counterfactual dataset, getting rid of expensive network re-training or bias labeling. Our approach inherently applies to in-processing debiasing.

- Extensive experiments and detailed analysis on multiple datasets demonstrate that our framework can obtain competing accuracies with significantly smaller biases and much fewer data and computational costs.

## 2 Related Works

### 2.1 Group, Individual and Counterfactual Fairness

The pursuit of fairness in machine learning has led to the proposal of fairness-specific metrics. These metrics have been mainly categorized into two types: metrics for group fairness that require similar average outputs of different demographic groups [34–38]; and metrics for individual fairness that necessitate similarity in the probability distributions of individuals that are similar in respect to a specific task, regardless of their demographic group [39–42]. Generally, statistical parity among protected groups in each class (group fairness) could be intuitively unfair at the individual level [43]. Moreover, existing fairness metrics put a heavy emphasis on model predictions, while underestimating the significance of sensitive attributes for decision-making and are insufficient to explain the cause of unfairness in the task [31, 44]. Recently, [31] introduces counterfactual fairness, a causal approach to address individual fairness, which enforces that the distribution of potential predictions for an individual should remain consistent when the individual's protected attributes had been different in a causal sense. In contrast to existing individual bias metrics, counterfactual fairness can explicitly model the causality between biased attributes and unfair predictions, which provides explainability for different biases that may arise towards individuals based on sensitive attributes [45–47].

### 2.2 Bias Mitigation

Proposed debiasing mechanisms are typically categorized into three types[2, 13, 14]: pre-processing, in-processing, and post-processing. Pre- and in-processing algorithms account for fairness before and during the training process, where typical techniques entail dataset modification [15–19] and feature manipulation [20–24, 26, 25]. Post-processing algorithms are performed after training, intending to achieve fairness without the need of modifying data or re-training the model. Current post-processing algorithms make more fair decisions by tweaking the output scores [48–50]. For instance, Hardt [27] achieves equal odds or equal opportunity by flipping certain decisions of the classifier according to their sub-groups. [29, 28] select different thresholds for each group, in a manner that maximizes accuracy and minimizes demographic parity. However, achieving group fairness by simply changing the predictions of several individuals is questionable, e.g., the process might be unfair to the selected individuals, leading to an unsatisfactory trade-off between accuracy and fairness.

### 2.3 Machine Unlearning

Machine unlearning [51–53] is a new paradigm to forget a specific data sample and remove its corresponding influence from a trained model, without the requirement to re-train the model from scratch. It fulfills a user's right to unlearn her private information, i.e., the right to be forgotten, in accordance with requests from the General Data Protection Regulation (GDPR) [54]. Existing unlearning approaches can be roughly categorized into two types: exact unlearning [55, 56] and approximate unlearning [57–60]. Data influence-based unlearning is a representative branch of approximate unlearning that utilizes influence functions [32] to approximate and remove the effect of a training sample on the model's parameters [61–63]. In this paper, we are inspired by the paradigm of machine unlearning and extend it to remove the model's bias from a deep network without retraining it from scratch.

# 3 Method

## 3.1 Overview and Preliminaries

**Problem Formulation.** Consider a supervised prediction task with fairness considerations that maps input attributes $\mathcal{A}$ (biased attribute) and $\mathcal{X}$ (other attributes except $\mathcal{A}$) to certain outputs $\mathcal{Y}$ (labels). The training dataset $D_{tr}$ can be represented as $\{z_1, z_2, ..., z_n\}$ where each training point $z_i = \{(a_i, x_i),\ y_i\} \in \mathcal{A} \times \mathcal{X} \times \mathcal{Y}$. Let $f_{\hat{\theta}}$ denote the trained model (predictor) with parameter $\hat{\theta}$. Let $L(z_i, \theta)$ denote the loss on the training sample $z_i$ w.r.t. parameter $\theta$. It is deemed biased if a *biased* attribute $a$ is highly correlated but wrongly correlated to the prediction $\hat{y} = f_{\hat{\theta}}(x, a)$, e.g., a CNN is biased if it predicts hair color (black/blonde) with the biased attribute *genders* (male/female).

**Motivation.** In large part, existing works focused on measuring fairness with implicit quantitative values (e.g. accuracy). However, they do not provide explicit illustrations on whether the decision-making is based on sensitive/protected attributes. Furthermore, based on the bias identified, research on how such bias is learned from training samples is limited. Our proposed method bridges this gap with two components: identifying bias from different predictions with counterfactual samples and evaluating the biased-effect from training samples with a modified influence function. Furthermore, we propose a novel machine unlearning-based method to efficiently and effectively remove the biases.

**Counterfactual Fairness.** We identify the biases of trained models with the concept of counterfactual fairness [31, 46, 45] which better models the causality between biased attributes and unfair predictions. We detail the definition following [31]:

**Definition 1** (Counterfactual fairness). *A trained model $f_{\hat{\theta}}$ is counterfactual fair on $\mathcal{A}$ if for any $a, \bar{a} \in \mathcal{A}$,*

$$P(\hat{Y}_{A \leftarrow a} = y \mid X = x, A = a) = P(\hat{Y}_{A \leftarrow \bar{a}} = y \mid X = x, A = a), \tag{1}$$

*for all $x \in \mathcal{X}$ attainable by $X$.*

Note that $y = f_{\bar{\theta}}(X, A)$, which implies the process of attribute changing. The definition suggests that, for any individual, changing $a$, i.e., from $a$ to $\bar{a}$, while holding other attributes $x$ unchanged should not change the distribution of $\hat{Y}$ if $a$ is a biased attribute.

**Influence function.** Influence functions, a standard technique from robust statistics, are recently extended to characterize the contribution of a given training sample to predictions in deep networks [32, 64, 65], e.g., identify whether a sample is helpful or harmful for model predictions. A popular implementation of influence functions is to approximate the effects by applying the perturbation $z = (x,\ y) \mapsto z_\delta = (x + \delta,\ y)$ [32] that define the parameters resulting from moving $\epsilon$ mass from $z$ onto $z_\delta$: $\hat{\theta}_{\epsilon, z_\delta, -z} = \arg\min_{\theta \in \Theta} \frac{1}{n} \sum_{i=1}^{n} L(z_i, \theta) + \epsilon L(z_\delta, \theta) - \epsilon L(z, \theta)$. An approximated computation of the influence as in [32] can be defined as:

$$\left. \frac{d\hat{\theta}_{\epsilon, z_\delta, -z}}{d\epsilon} \right|_{\epsilon=0} = -H_{\hat{\theta}}^{-1} \big( \nabla_\theta L(z_\delta, \hat{\theta}) - \nabla_\theta L(z, \hat{\theta}) \big). \tag{2}$$

## 3.2 Bias Identification and Biased-Effect Evaluation

**Counterfactual bias identification.** We first identify the biases in a trained model with counterfactual concepts. Given a trained model $f_{\hat{\theta}}$ and an attribute of interest $\mathcal{A}$, a primary question is whether $f_{\hat{\theta}}$ is fair on $\mathcal{A}$. We employ an external dataset $D_{ex}$ (can be constructed from the test set) to identify biases. To measure how prediction changes in accordance with the attribute, for each sample $c_i = (x_i, a_i) \in D_{ex}$, where $a_i \in \mathcal{A}$, we alter $a_i$ while keeping $x_i$ unchanged based on the requirements of counterfactual fairness. The generated counterfactual sample is denoted as $\bar{c}_i = (x_i, \bar{a}_i), \bar{a}_i \in \mathcal{A}$. We further define the counterfactual bias of the model $f_{\hat{\theta}}$ on sample $c_i$ as the difference in predictions:

$$B(c_i, \mathcal{A}, \hat{\theta}) = \left| P(\hat{Y} = f_{\hat{\theta}}(X, A)) \mid X = x_i, A = a_i)) - P(\hat{Y} = f_{\hat{\theta}}(X, A) \mid X = x_i, A = \bar{a}_i) \right|. \tag{3}$$

The counterfactual bias on the whole dataset $D_{ex}$ can be represented as the average of individual counterfactual biases:

$$B(D_{ex}, \mathcal{A}, \hat{\theta}) = \frac{1}{|D_{ex}|} \sum_i B(c_i, \mathcal{A}, \hat{\theta}). \tag{4}$$

The measured bias is a scalar normalized from 0 to 1. We set a bias threshold $\delta$ that if the measured $B(D_{ex}, \mathcal{A}, f_{\hat{\theta}})$ is larger than $\delta$, we regard $f_{\hat{\theta}}$ to be biased on $\mathcal{A}$. Note that our method could also generalize to other individual bias metrics besides Eq. 3.

**Biased-Effect Evaluation.** Based on the identified counterfactual bias, we then investigate how the bias on $\mathcal{A}$ is learned by the model from training samples. Considering $B(\hat{\theta})$ measured on any $\mathcal{A}$ with any $D_{ex}$, our goal is to quantify how each training point $z_k$ in the training set $D_{tr}$ contributes to $B(\hat{\theta})$. Let's denote the empirical risk minimizer as $\hat{\theta} = \arg\min_\theta \frac{1}{n} \sum_{i=1}^n L(z_i, \theta)$, and assume that the empirical risk is twice-differentiable and strictly convex in $\theta$. The influence function [64] provides an approximation on the updates to parameters if $z_k$ were removed from $D_{tr}$ with a small coefficient $\epsilon$. The new parameters can be obtained as $\hat{\theta}_{\epsilon, z_k} = \arg\min_\theta \frac{1}{n} \sum_{i=1, i\neq k}^n L(z_i, \theta) + \epsilon L(z_k, \theta)$. By doing so, the influence of removing $z_k$ on the bias $B(\hat{\theta})$ can be defined as:

$$I_{up,bias}(z_k, B(\hat{\theta})) = \frac{dB(\hat{\theta}_{\epsilon, z_k})}{d\epsilon}\bigg|_{\epsilon=0} = \frac{dB(\hat{\theta}_{\epsilon, z_k})}{d\hat{\theta}_{\epsilon, z_k}} \frac{d\hat{\theta}_{\epsilon, z_k}}{d\epsilon}\bigg|_{\epsilon=0} = -\nabla_{\hat{\theta}} B(\hat{\theta}) H_{\hat{\theta}}^{-1} \nabla_{\hat{\theta}} L(z_k, \hat{\theta}), \tag{5}$$

where $H_{\hat{\theta}} \overset{\text{def}}{=} \frac{1}{n} \sum_{i=1}^n \nabla_\theta^2 L(z_k, \hat{\theta})$ is the positive definite (PD) Hessian, and the closed form expression of $\frac{d\hat{\theta}_{\epsilon, z_k}}{d\epsilon}\big|_{\epsilon=0}$, explaining the influence of $z_k$ to model parameters, is provided by the influence function [32]. Note that "up" denotes "upweight". Refer to Appendix A for the derivation. Intuitively, this equation can be understood in two parts: the latter part calculates the impact of removing on the parameters. The former part corresponds to the derivative of bias with respect to parameters, assessing how changes in parameters affect the bias. Hence, this equation quantifies the influence of removing on the bias. Note that $B(\hat{\theta})$ can be any bias measurement of interest. Taking $B(D_{ex}, \mathcal{A}, \hat{\theta})$ defined in Eq. 4 as an example, the influence on counterfactual bias can be boiled down as:

$$I_{up,bias}(z_k, B(D_{ex}, \mathcal{A}, \hat{\theta})) = \frac{1}{|D_{ex}|} \sum_{c_i \in D_{ex}} (\nabla_{\hat{\theta}} f_{\hat{\theta}}(c_i) - \nabla_{\hat{\theta}} f_{\hat{\theta}}(\bar{c}_i)) H_{\hat{\theta}}^{-1} \nabla_{\hat{\theta}} L(z_k, \hat{\theta}), \tag{6}$$

where $I_{up,bias}(z_k, B)$ is a scalar that measures how each training sample contributes to $B$. If removing the point $z_k$ increases the bias, we regard $z_k$ as a helpful sample, or harmful otherwise. We provide an illustration of the helpful and harmful samples with a toy example in Section B.1.

### 3.3 Bias Removal via Machine Unlearning

After quantifying how biases are learned by the model from harmful samples, the next question is how to remove such biases. Here we propose a machine unlearning-based strategy to remove the biases caused by harmful samples. In particular, we exploit the powerful capability of machine unlearning paradigms for forgetting certain training samples [66, 62, 63, 61]. Specifically, for a bias measurement $B(\hat{\theta})$, we first rank the influence $I_{up,bias}(z_k, B(\hat{\theta}))$ of every training sample $z_k$ in $D_{tr}$, and then select the top-$K$ harmful samples. Afterward, we unlearn, i.e., let the model forget, these samples by updating the model parameters $\theta$ with a Newton update step as in [63]:

$$\theta_{new} = \hat{\theta} + \sum_{k=1}^K H_{\hat{\theta}}^{-1} \nabla_{\hat{\theta}} L(z_k, \hat{\theta}), \tag{7}$$

where $H_{\hat{\theta}}^{-1} \nabla_{\hat{\theta}} L(z_k, \hat{\theta}) = I_{up,params}(z_k)$ is explained as the influence of $z_k$ on model parameter [32]. Note that $I_{up,params}(z_k)$ share similar computation as in Eq. 6, while $I_{up,params}(z_k)$ estimates the influence on model parameter and $I_{up,bias}(z_k, B)$ focuses on influence on biases.

Our unlearning strategy is further refined following the observations from experiments in Section B.1. In particular, by ranking and visualizing the harmful and helpful samples on the biases (as shown in Fig. 5), we have observed that the harmful samples heavily lead to biased/error correlations (i.e., bias-aligned) while the helpful samples behave oppositely (i.e., bias-conflicting). Hence, we propose a straightforward solution that further mitigates the influence of a harmful sample with a

bias-conflicting sample. Consequently, we update the parameters to unlearn the harmful samples by:

$$\theta_{new} = \hat{\theta} + \sum_{k=1}^{K} H_{\hat{\theta}}^{-1}(\nabla_{\hat{\theta}} L(z_k, \hat{\theta}) - \nabla_{\hat{\theta}} L(\bar{z}_k, \hat{\theta})), \qquad (8)$$

where $\bar{z}_k$ denotes the bias-conflicting sample of $z_k$. Following the explanation in influence theory [32], our unlearn mechanism removes the effect of perturbing a training point $(\bar{a}, x, y)$ to $(a, x, y)$. In other words, we not only remove the influence caused by harmful sample $z_k$, but further ensure fairness with the corresponding counterfactual sample $\bar{z}_k$, see more details in Section B.1, 4.4 and Appendix.

**Alternative Efficient Unlearn with Cheap External Datasets.** In the above sections, the unlearning process is based on the assumption that we could access the original training sample $z_k$ to identify and evaluate biases and then forget them. However, in practice, the training set might be too large or even unavailable in the unlearning phase. In response, we further propose to approximate the unlearning mechanism with a small external dataset. As the influence to be removed can be obtained from the change of the protected attribute, we can construct the same modification to the protected attribute on external samples. In particular, we employ the $D_{ex}$ as in Section 3.2 to construct counterfactual pairs for unlearning, which redefines Eq. 21 as:

$$\theta_{new} = \hat{\theta} + \sum_{i} H_{\hat{\theta}}^{-1}(\nabla_{\hat{\theta}} L(c_i, \hat{\theta}) - \nabla_{\hat{\theta}} L(\bar{c}_i, \hat{\theta})). \qquad (9)$$

As $D_{ex}$ can be easily obtained from an external dataset rather than the training set, the practical applicability of our method could be greatly enhanced, as demonstrated in the experiments.

### 3.4 Model Generalization

**Extension to Different Biases.** To fulfill different fairness demands, we further discuss the generalization of the bias function $B(\hat{\theta})$ in Eq. 6 to other bias measurements. We provide the extension to the most frequently used group fairness measurement demographic parity [34] which requires equal positive prediction assignment across subgroups (e.g. male and female). Eq. 6 can be rewritten as:

$$I_{up,bias}(z_k) = -(\nabla_{\hat{\theta}} \frac{1}{|\mathcal{G}_{A=1}|} \sum_{c_i \in \mathcal{G}_{A=1}} f_{\hat{\theta}}(c_i) - \nabla_{\hat{\theta}} \frac{1}{|\mathcal{G}_{A=0}|} \sum_{c_j \in \mathcal{G}_{A=0}} f_{\hat{\theta}}(c_j)) H_{\hat{\theta}}^{-1} \nabla_{\hat{\theta}} L(z_k, \hat{\theta}), \quad (10)$$

where $\mathcal{G}_{A=1}$ and $\mathcal{G}_{A=0}$ represents the subgroup with protected attribute $A = 1$ and $0$. The extension to equal opportunity [35], which requires the positive predictions to be equally assigned across positive classes, can be rewritten as:

$$I_{up,bias}(z_k) = -(\nabla_{\hat{\theta}} \frac{1}{|\mathcal{G}_{1,1}|} \sum_{c_i \in \mathcal{G}_{1,1}} f_{\hat{\theta}}(c_i) - \nabla_{\hat{\theta}} \frac{1}{|\mathcal{G}_{0,1}|} \sum_{c_j \in \mathcal{G}_{0,1}} f_{\hat{\theta}}(c_j)) H_{\hat{\theta}}^{-1} \nabla_{\hat{\theta}} L(z_k, \hat{\theta}), \quad (11)$$

where $\mathcal{G}_{1,1}$ represents the sub-group where $A = 1$ and $Y = 1$.

**Extension to Deep Models.** In the previous sections, it's assumed that $\hat{\theta}$ could be the global minimum. However, if $\hat{\theta}$ is obtained in deep networks trained with SGD in a non-convex setting, it might be a local optimum and the exact influence can hardly be computed. We follow the strategy in [32] to approximate the influence in deep networks, and empirically demonstrate the effectiveness of FMD in deep models. Moreover, for deep networks where a linear classifier is stacked on a backbone feature extractor, we apply our unlearning mechanism to the linear classifier or several top MLP layers.

**Efficient Influence Computation.** A critical challenge to compute the influence in Eq. 6 is to explicitly calculate the inverse Hessian. Here we employ the implicit Hessian-vector products (HVPs) [32, 67] to efficiently approximate $\nabla_{\hat{\theta}} L(z_k, \hat{\theta})$. Meanwhile, $\nabla_{\hat{\theta}} L(z_k, \hat{\theta})$ in Eq. 6 can be pre-calculated and applied to different $\nabla_{\hat{\theta}} B(\hat{\theta})$. To avoid the $O(d^3)$ computational cost to calculate the inverse Hessian in every step, we pre-calculate it before the removal and keep it constant during unlearning phase [63]. The alternative strategy which continuously updates the inversion Hessian is also analyzed in the Appendix.

---

**Algorithm 1:** The FMD framework.

**Input:** dataset $D_{ex}$, loss $\mathcal{L}$, attribute of interest $\mathcal{A}$, Hessian matrix $H$, bias threshold $\delta$, parameter $\theta$, $n = \|D_{ex}\|$.
$B \leftarrow B(D_{ex}, \mathcal{A}, \hat{\theta})$
$H^{-1} \leftarrow \text{Inverse}(H)$
**if** $B > \delta$ **then**
    **for** $i = 1,2,3,...,n$ **do**
        $\triangle \leftarrow \nabla_{\hat{\theta}} L(c_i, \hat{\theta}) - \nabla_{\hat{\theta}} L(\bar{c}_i, \hat{\theta})$
        $\theta \leftarrow \theta + H^{-1}\triangle$
    **end**
**end**
**Output:** $\theta$

# 4 Experiment

## 4.1 Experiment details

**Dataset.** Our experiments are conducted on three datasets. **Colored MNIST** is constructed by adding color bias to the MNIST dataset [68]. Bias-aligned samples are constructed by adding a particular color to a particular digit like {Digit1_Color1} while other colors are for bias-conflicting samples. Following [3, 69, 70], we build 3 different training sets by setting different biased ratios {0.995, 0.99, 0.95} for biased-aligned training samples, where a high ratio indicates a high degree of bias. **CelebA** [71] is a face recognition dataset with 40 types of attributes like gender, age (young or not), and lots of facial characteristics (such as hair color, smile, beard). We choose Gender as the bias attribute, and Blonde hair and Attractive as the outputs following [7, 8]. **Adult Income Dataset** is a publicly available dataset in the UCI repository [72] based on the 1994 U.S. census data. The dataset records an individual's income (more or less than $50,000 per year) along with features such as occupation, marital status, and education. In our experiment, we choose gender and race as biased attributes following [73, 74]. We follow the pre-processing procedures in [75]. As for the experiment on the language model, we use **StereoSet** [76] as our test set. StereoSet is a large-scale natural dataset to measure stereotypical biases in gender, profession, race, and religion.

**Baselines.** For the sanity check experiment on a toy Colored MNIST dataset, we use a vanilla logistic regression model as the baseline. For experiments with deep networks, we compare our method with one pre-processing baseline Reweigh [77], 6 in-processing debiasing baselines (LDR [25], LfF [78], Rebias [79], DRO [7], SenSEI [80], and SenSR [81]) and 4 post-processing baselines (EqOdd [35], CEqOdd [35], Reject [82] and PP-IF [83]). We compare our method on language model with five debiasing baselines: Counterfactual Data Augmentation (CDA) [84], Dropout [85], Iterative Null-space Projection (INLP) [86], Self-debias [87], and SentenceDebias [88]. Details can be referred to Appendix C.

**Construction of Counterfactual Dataset** $D_{ex}$**.** We separately construct counterfactual sets for the three datasets, while bias-aligned samples in the small dataset $D_{ex}$ are all split from the test set. In the Colored MNIST dataset, we randomly add another color on the same digit image as the counterfactual sample. As for the Adult dataset, we flip the protected attribute to the opposite while keeping other attributes and target labels exactly the same. For the CelebA dataset, we select images with the same target labels but the opposite protected attribute. To fulfill the request for counterfactual, we rank the similarity between images by comparing the overlap of other attributes and choose the most similar pair to form the factual and counterfactual samples. Part of the generated sample pairs is visualized in Fig. 2. Note that for the CelebA dataset, the counterfactual data are not that strict as the gender attribute is not independent of other features in the natural human facial images. We use Crows-Pairs [89] as our external dataset for the language model. Each sample in Crows-Pairs consists of two sentences: one that is more stereotyping and another that is less stereotyping, which can be utilized as counterfactual pairs.

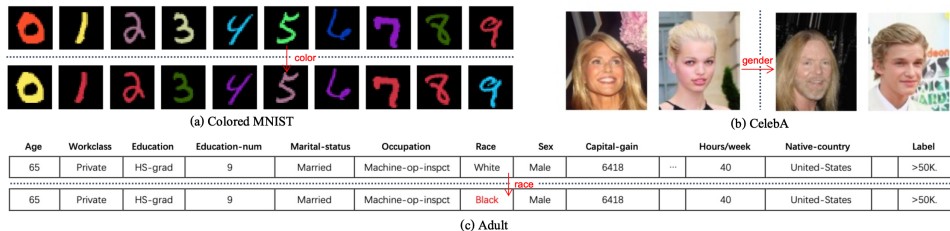

Figure 2: Visualization of factual and counterfactual pairs for three datasets.

**Implementation details.** We use multi-layer perceptron (MLP) with three hidden layers for Colored MNIST and Adult, and ResNet-18 [90] for CelebA following the setting in [8]. During training, we set the batch size of 256 for Colored MNIST and Adult, respectively, and 64 for CelebA following [25, 78, 7]. We use pre-trained BERT [91] and GPT-2 [92], provided by Huggingface. During unlearning, we freeze the parameters of all other layers except the last classifier layer. The running time of all baselines is evaluated on a single RTX3090 GPU for a fair comparison. In our experiment, we select the number of samples k=5000 for Colored MNIST, and k=200 for both Adult and CelebA. The bias threshold is set to 0.

## 4.2 Sanity Check on Logistic Regression with a Toy Dataset

We conduct an experiment on a logistic regression task to illustrate our method. We simplify the Colored MNIST classification task to a binary classification problem of distinguishing between only digits 3 and 8, on a training set with a bias ratio of 0.95. and a balanced test set. We trained a regularized logistic regressor: $\operatorname{argmin}_{w \in R^d} \sum_{i=1}^{n} l(w^T x_i, y_i) + \lambda \|w\|_2^2$. Fig. 5(a) illustrates the classification results of the vanilla regressor on part of test samples. We denote Digit by shape (triangle and rectangle) and Color by color (red and blue). The solid line represents the learned classification boundary and the dotted line represents the expected classification boundary. The test accuracy is $0.6517$ and it can be observed that most bias-conflict samples tend to be misclassified according to their colors. Moreover, we select and visualize the most helpful and harmful samples in Fig. 5(c) based on Eq. 6. We found that the most helpful samples are in the 5% bias-conflict samples while harmful samples are bias-aligned samples. The unlearning curve is provided in Fig. 5(b). With only 50 samples, the accuracy is improved amazingly by 25.71% and the counterfactual bias decreases by 0.2755, demonstrating the effectiveness of our method.

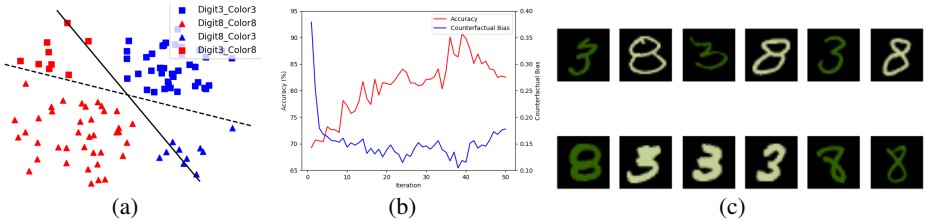

Figure 3: (a) Illustration of the learned pattern on our toy dataset.(b) Accuracy and bias curves during unlearning. (b) Visualization of helpful samples (top row) and harmful samples (bottom row).

## 4.3 Experiment on Deep Models

**Results on Colored MNIST.** Tab. 1 shows the comparisons on the Colored MNIST dataset. We reported test accuracy, counterfactual bias, debiasing time, and the number of samples used for all methods. Our approach demonstrates competing performance on accuracy and superior performance on bias compared with retraining baselines. Meanwhile, we only make use of one-tenth of unlearning samples and reduce the debiasing time by 1-2 magnitudes.

| Bias Ratio | Method | Acc.(%) ↑ | Bias ↓ | Time(s) | # Samp. |
|---|---|---|---|---|---|
| | Vanilla | 38.59 | 0.5863 | - | - |
| | LDR | 66.76 | 0.4144 | 1,261 | 50 k |
| 0.995 | LfF | 56.45 | 0.3675 | 661 | 50 k |
| | Rebias | 71.24 | 0.3428 | 1,799 | 50 k |
| | Ours | **71.70** | **0.3027** | **59** | 5 k |
| | Vanilla | 51.34 | 0.4931 | - | - |
| | LDR | 76.48 | 0.2511 | 1,330 | 50 k |
| 0.99 | LfF | 64.71 | 0.2366 | 726 | 50 k |
| | Rebias | **80.41** | 0.2302 | 1,658 | 50 k |
| | Ours | 80.04 | **0.2042** | **48** | 5 k |
| | Vanilla | 77.63 | 0.2589 | - | - |
| | LDR | **90.42** | 0.2334 | 1,180 | 50 k |
| 0.95 | LfF | 85.55 | 0.1264 | 724 | 50 k |
| | Rebias | 89.63 | 0.1205 | 1,714 | 50 k |
| | Ours | 89.26 | **0.1189** | **56** | 5 k |

Table 1: Results on Colored MNIST. (**bold**: best performance, underline: second best performance.)

| Attr. | Method | Acc.(%) ↑ | Bias ↓ | Time(s) | # Samp. |
|---|---|---|---|---|---|
| | Vanilla | **85.40** | 0.0195 | - | - |
| | LDR | 77.69 | 0.0055 | 927 | 26,904 |
| | LfF | 73.08 | 0.0036 | 525 | 26,904 |
| Gender | Rebias | 76.57 | 0.0041 | 1292 | 26,904 |
| | Reweigh | 82.60 | 0.0051 | 36 | 26,904 |
| | SenSR | 84.09 | 0.0049 | 571 | 26,904 |
| | SenSeI | 83.91 | 0.0016 | 692 | 26,904 |
| | PP-IF | 81.96 | 0.0027 | 13 | 26,904 |
| | Ours | 81.89 | **0.0005** | **2.49** | 500 |
| | Vanilla | **84.57** | 0.0089 | - | - |
| | LDR | 78.32 | 0.0046 | 961 | 26,904 |
| | LfF | 75.16 | 0.0024 | 501 | 26,904 |
| Race | Rebias | 77.89 | 0.0038 | 1304 | 26,904 |
| | Reweigh | 82.97 | 0.0015 | 36 | 26,904 |
| | SenSR | 84.09 | 0.0036 | 571 | 26,904 |
| | SenSeI | 83.91 | 0.0015 | 692 | 26,904 |
| | PP-IF | 82.37 | 0.0015 | 13 | 26,904 |
| | Ours | 83.80 | **0.0013** | **2.54** | 500 |

Table 2: Results on Adult.

**Results on Adult.** The results are in Table 2. It can be observed that the vanilla method performed the best in accuracy on both tasks, since in the real-world dataset, race and gender are biased w.r.t income in both training and test set and the well-trained model fits this correlation. However, to achieve fair prediction, we would not expect biased attributes to dominate predictions. Compared with other debiasing methods, our method achieved the best results in both accuracy and bias, with much less debiasing time on a smaller dataset.

**Results on CelebA.** We compare on average accuracy (Avg.), Unbiased accuracy [8] (Unb.) tested on the balanced test set, and Worst-group accuracy [7] (Wor.) tested on the unprivileged group to illustrate the performance, as reported in Tab. 3. It can be observed that the vanilla model performs well on the whole dataset (Avg.) but scores a really low accuracy (Wor.) on the worst group, which means the learned model heavily relies on the bias attribute to achieve high accuracy. Our method obviously bridges this gap and outperforms all other debiasing baselines on Wor. and Unb. in the two experiments. The experiments also demonstrate that our method is consistently feasible even in the absence of perfect standardized counterfactual samples in real-world datasets, by selecting a certain amount of approximate counterfactual data.

**Results on Large Language Models (LLM).** We further extended our method to the LLM debiasing scenario. Results are presented in Tab. 4. We report two metrics: Language Modeling Score (LMS) measures the percentage of instances in which a language model prefers the meaningful over meaningless association. The LMS of an ideal language model is 100 (the higher the better). Stereotype Score

| Attr. | Method | Unb.(%) ↑ | Wor.(%) ↑ | Avg.(%) ↑ | Bias ↓ | Time(s) |
|---|---|---|---|---|---|---|
| Blonde | Vanilla | 66.27 | 47.36 | **94.90** | 0.4211 | - |
| | LfF | 84.33 | 81.24 | 93.52 | 0.2557 | 67,620 |
| | LDR | 85.01 | 82.32 | 86.67 | 0.3126 | 24,180 |
| | DRO | 85.66 | 84.36 | 92.90 | 0.3206 | 28,860 |
| | Ours | **89.73** | **87.15** | 93.41 | **0.0717** | **191** |
| Attractive | Vanilla | 63.17 | 40.59 | 77.42 | 0.3695 | - |
| | LfF | 67.44 | 52.25 | 77.24 | 0.2815 | 67,560 |
| | LDR | 68.14 | 54.47 | **81.70** | 0.2986 | 24,420 |
| | DRO | 66.14 | 62.33 | 78.35 | 0.3004 | 30,540 |
| | Ours | **72.18** | **68.16** | 80.99 | **0.1273** | **187** |

Table 3: Results on CelebA.

(SS) measures the percentage of examples in which a model prefers a stereotypical association over an anti-stereotypical association. The SS of an ideal language model is 50 (the closer to 50 the better). It shows that our method can outperform or achieve comparable performance with baseline methods. As for BERT, our method reaches the best (denoted by **bold**) or second best (denoted by underline) performance in 5 of 6 metrics. Description of baselines can be referred to Appendix. C.2.

| Backbone | Attribute | Method | SS | LMS | Attribute | Method | SS | LMS | Attribute | Method | SS | LMS |
|---|---|---|---|---|---|---|---|---|---|---|---|---|
| BERT | gender | Vanilla | 60.28 | 84.17 | race | Vanilla | 57.03 | 84.17 | religion | Vanilla | 59.7 | 84.17 |
| | | CDA | 59.61 | 83.08 | | CDA | 56.73 | 83.41 | | CDA | 58.37 | 83.24 |
| | | Dropout | 60.66 | 83.04 | | Dropout | 57.07 | 83.04 | | Dropout | 59.13 | 83.04 |
| | | INLP | **57.25** | 80.63 | | INLP | 57.29 | 83.12 | | INLP | 60.31 | 83.36 |
| | | Self-debias | 59.34 | 84.09 | | Self-debias | 54.30 | 84.24 | | Self-debias | **57.26** | 84.23 |
| | | SentDebias | 59.37 | 84.20 | | SentDebias | **57.78** | 83.95 | | SentDebias | 58.73 | 84.26 |
| | | Ours | 57.77 | **85.45** | | Ours | 57.24 | **84.19** | | Ours | 57.85 | **84.90** |
| GPT-2 | gender | Vanilla | 62.65 | 91.01 | race | Vanilla | 58.9 | 91.01 | religion | Vanilla | 63.26 | 91.01 |
| | | CDA | 64.02 | 90.36 | | CDA | 57.31 | 90.36 | | CDA | 63.55 | 90.36 |
| | | Dropout | 63.35 | 90.40 | | Dropout | 57.50 | 90.40 | | Dropout | 64.17 | 90.40 |
| | | INLP | 60.17 | **91.62** | | INLP | 58.96 | 91.06 | | INLP | 63.95 | **91.17** |
| | | Self-debias | 60.84 | 89.07 | | Self-debias | 57.33 | 89.53 | | Self-debias | 60.45 | 89.36 |
| | | SentDebias | **56.05** | 87.43 | | SentDebias | **56.43** | **91.38** | | SentDebias | 59.62 | 90.53 |
| | | Ours | 60.42 | 91.01 | | Ours | 60.42 | 91.01 | | Ours | **58.43** | 86.13 |

Table 4: Results with Large Language Models (BERT and GPT-2).

## 4.4 Analysis

**Effectiveness on Different Bias Metrics.** We validate the generalization ability of our unlearning method based on different fairness metrics on the Colored MNIST with bias severity 0.99. In Tab. 5, we compare the performance of unlearning harmful samples based on three different biases: Counterfactual bias (Co.), Demographic parity bias (De.) [34], and Equal opportunity bias (Eo.) [35]. For each experiment, we report the changes in three biases. We can note that our method is consistently effective on all three bias metrics. Meanwhile, our counterfactual-based unlearning can significantly outperform the other two in terms of accuracy, Co., and De., and is comparable with them on Eo..

| | Acc.(%) ↑ | Co. ↓ | De. ↓ | Eo. ↓ |
|---|---|---|---|---|
| Vanilla | 65.17 | 0.3735 | 0.5895 | 0.2235 |
| Unlearn by De. | 71.52 | 0.1796 | 0.4026 | 0.0116 |
| Unlearn by Eo. | 71.12 | 0.1826 | 0.4217 | **0.0103** |
| Unlearn by Co. (Ours) | **87.90** | **0.1051** | **0.1498** | 0.0108 |

Table 5: Ablation on Different Biases.

| | Acc. ↑ | Bias ↓ | Time(s)↓ |
|---|---|---|---|
| Vanilla | 65.17 | 0.3735 | - |
| Unlearn by Eq. 7 | 90.68 | 0.1182 | 36.87 |
| Unlearn by Eq. 8 | **91.18** | **0.1023** | 39.63 |
| Unlearn by Eq. 9 (Ours) | 90.42 | 0.1051 | **0.059** |

Table 6: Ablation on Unlearning Strategies.

**Effectiveness of Unlearn Strategies.** We empirically investigate the feasibility of the unlearning mechanism on training and external samples on the Colored MNIST with bias severity 0.99. In Tab. 6, we report the results of unlearning harmful training samples (Eq. 7), unlearning by replacing harmful

samples with their bias-conflict helpful samples (Eq. 21) and unlearning with external counterfactual sample pairs (Eq. 23). It can be observed that unlearning in the training dataset can achieve higher accuracy and less bias, and Eq. 21 excels on both metrics. But unlearning with training samples requires much more time and training samples might not be available in practice, while unlearning with external samples provides a satisfactory alternative.

| Attr. | Method | Acc.(%) ↑ | Co. ↓ | De. ↓ | Eq. ↓ | Time(s) |
|---|---|---|---|---|---|---|
| Gender | EqOdd | 82.71 | 0.0247 | 0.5991 | **0.0021** | **0.0113** |
| | CEqOdd | 83.22 | 0.0047 | 0.4469 | 0.0125 | 0.5583 |
| | Reject | 74.63 | 0.0876 | 0.2744 | 0.3140 | 14.420 |
| | Ours | **83.49** | **0.0019** | **0.1438** | 0.0460 | 0.0389 |
| Race | EqOdd | 83.22 | 0.0139 | 0.7288 | **0.0021** | **0.0105** |
| | CEqOdd | 82.88 | 0.0012 | 0.6803 | 0.0054 | 3.6850 |
| | Reject | 74.63 | 0.1156 | 0.4349 | 0.1825 | 14.290 |
| | Ours | **83.12** | **0.0006** | **0.4219** | 0.0367 | 0.0360 |

Table 7: Discussion on Post-processing Methods.

| Method | # Lay. | # Para. | Acc(%) ↑ | Bias ↓ | Time(s) |
|---|---|---|---|---|---|
| Vanilla | - | - | 51.34 | 0.4931 | - |
| Ours[1] | 1 | 1 K | 71.19 | **0.2757** | **3.75** |
| Ours[2] | 2 | 11 K | **74.18** | 0.3134 | 432.86 |
| Ours[3] | 3 | 21 K | 61.45 | 0.2949 | 496.44 |

Table 8: Ablation on # MLP Layers.

**Discussion on Post-processing Methods.** We compare our method to post-processing methods, i.e., Equalized Odds Post-processing (EqOdd) [35], Calibrated Equalized Odds Post-processing (CEqOdd) [35] and Reject Option Classification (Reject) [82], as shown in Tab. 7. Note these methods only apply to logistic regression. Our method outperforms them in most cases on Adult. It is also worth noting that these post-processing methods aimed at a specific fairness measure tend to exacerbate unfairness under other measures while our method consistently improves the fairness under different measures.

**Ablation on the Number of Samples.** Fig. 4 demonstrates the sensitivity of our unlearning performance w.r.t. number of samples on Colored MNIST with a bias ratio of 0.99. The accuracy increases and bias decreases incrementally with more samples, and becomes steady after the number is beyond 5,000. On the other hand, the unlearning time increases linearly with the number of samples. Additionally, constructing a large number of counterfactual samples in practice might be time-consuming as well. Practical usage of the FMD would require a trade-off based on utility requirements.

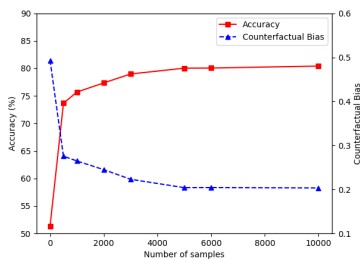

Figure 4: Ablation on # Samples.

**Ablation on Number of Fine-tuning Layers.** We explore the impact of unlearning different numbers of layers (i.e., the last (one), two, three MLP) on the Color MNIST, with results in Tab. 11. Interestingly, the accuracy excels with two layers but decreases with three layers. Additionally, fine-tuning multiple layers takes much longer time on computation on much more parameters. It is also worth noting that our method could achieve such superior or competing performance even only by updating the last layer in deep models, which calls for more in-depth analysis in the future.

## 5 Conclusion and Limitation

Biased behaviors in contemporary well-trained deep neural networks can perpetuate social biases, and also pose challenges to the models' robustness. In response, we present FDM, an all-inclusive framework for fast and effective model debiasing. We explicitly measure the influence of training samples on bias measurement and propose a removal mechanism for model debiasing. Comprehensive experiments on multiple datasets demonstrate that our method can achieve superior/competing accuracies with a significantly lower bias as well as computational cost.

Our work preliminarily explored the application of our method to large language models, as well as more analysis on model fairness from different perspectives, which will be in our future plan. In addition, our method is not applicable to black-box models, which are of high interest in real-world scenarios. Our proposed method requires generating counterfactual pairs with labeled sensitive attributes, while many datasets do not have enough labels. Research on fairness with few/no attribute labels is still in the infant stage [93], and we will further explore it.

## Acknowledgements

This work is supported by the National Natural Science Foundation of China (Grant No. 62106222), the Natural Science Foundation of Zhejiang Province, China(Grant No. LZ23F020008) and the Zhejiang University-Angelalign Inc. R&D Center for Intelligent Healthcare.

## A    Influence Function on Bias and Extension to DNN.

### A.1    Deriving the Influence Function on Bias.

In this part, we provide detailed derivation of the influence function on bias in Eq. 5 in the main work. We first start from the influence function on parameters, we can also be referred to [32, 64].

Assuming there are n training samples $z_1, z_2..., z_n$, where $z_i = (x_i, y_i)$, and let $L(z, \theta)$ represent the loss function of sample $z$ under the model parameters $\theta$, then the trained $\hat{\theta}$ is given by:

$$\hat{\theta} = \text{argmin}_\theta R(\theta) = \text{argmin}_\theta \frac{1}{n} \sum_{i=1}^{n} L(z_i, \theta). \tag{12}$$

Study the impact of changing the weight of a training sample $z$ on the model parameters $\theta$. If we increase the weight of this sample $z$ in the training set by $\epsilon$, then the perturbed parameters $\hat{\theta}_{\epsilon,z}$ obtained according to ERM (Empirical Risk Minimization) will be:

$$\hat{\theta}_{\epsilon,z} = \arg \min_\theta R(\theta) + \epsilon L(z, \theta). \tag{13}$$

Define the parameter change $\Delta_\epsilon = \hat{\theta}_{\epsilon,z} - \hat{\theta}$, and note that, as $\hat{\theta}$ doesn't depend on $\epsilon$, the quantity we seek to compute can be written in terms of it:

$$\frac{d\hat{\theta}_{\epsilon,z}}{d\epsilon} = \frac{d\Delta_\epsilon}{d\epsilon}. \tag{14}$$

Since $\hat{\theta}_{\epsilon,z}$ is a minimizer of $R(\theta)$, therefore it satisfies the first-order derivative condition, which means the first-order derivative with respect to $\theta$ is zero:

$$0 = \nabla R(\hat{\theta}_{\epsilon,z}) + \epsilon \nabla L(z, \hat{\theta}_{\epsilon,z}). \tag{15}$$

Next, since $\hat{\theta}_{\epsilon,z} \to \hat{\theta}$ as $\epsilon \to 0$, we perform a Taylor expansion of the right-hand side:

$$0 \approx \{\nabla R(\hat{\theta}) + \epsilon \nabla L(z, \hat{\theta})\} + \nabla^2 R(\hat{\theta}) + \epsilon \nabla^2 L(z, \hat{\theta}) \Delta_\epsilon,$$

where we have dropped $o(\|\Delta_\epsilon\|)$ terms. Solving for $\Delta_\epsilon$, we get:

$$\Delta_\epsilon \approx - \{\nabla^2 R(\hat{\theta}) + \epsilon \nabla^2 L(z, \hat{\theta})\}^{-1} \{\nabla R(\hat{\theta}) + \epsilon \nabla L(z, \hat{\theta})\}.$$

Since $\hat{\theta}$ minimizes $R$, we have $\nabla R(\hat{\theta}) = 0$. Dropping $o(\epsilon)$ terms, we have

$$\Delta_\epsilon \approx - \nabla^2 R(\hat{\theta})^{-1} \nabla L(z, \hat{\theta}) \epsilon. \tag{16}$$

Note that it is assumed that $R$ is twice-differentiable and strongly convex in $\theta$. we define:

$$H_{\hat{\theta}} = \nabla^2 R(\hat{\theta}) = \frac{1}{n} \sum_{i=1}^{n} \nabla_\theta^2 L(z_i, \hat{\theta}) \tag{17}$$

exists and is positive definite. This guarantees the existence of $H_{\hat{\theta}}^{-1}$. The final influence function can be written as:

$$I_{up,params}(z) = \left.\frac{d\hat{\theta}_{\epsilon,z_k}}{d\epsilon}\right|_{\epsilon=0} = -H_{\hat{\theta}}^{-1}\nabla_{\hat{\theta}}L(z,\hat{\theta}), \tag{18}$$

Considering $B(\hat{\theta})$ measured on any $\mathcal{A}$ with any $D_{ex}$, our goal is to quantify how each training point $z$ in the training set $D_{tr}$ contributes to $B(\hat{\theta})$. We apply the chain rule on Eq. 18:

$$I_{up,bias}(z_k, B(\hat{\theta})) = \left.\frac{dB(\hat{\theta}_{\epsilon,z_k})}{d\hat{\theta}_{\epsilon,z_k}}\frac{d\hat{\theta}_{\epsilon,z_k}}{d\epsilon}\right|_{\epsilon=0} = -\nabla_{\hat{\theta}}B(\hat{\theta})H_{\hat{\theta}}^{-1}\nabla_{\hat{\theta}}L(z_k,\hat{\theta}), \tag{19}$$

Intuitively, this equation can be understood in two parts: the latter part calculates the impact of removing $z$ on the parameters. The former part corresponds to the derivative of bias with respect to parameters, assessing how changes in parameters affect the bias. Hence, this equation quantifies the influence of removing $z$ on the bias.

### A.2 Influence at Non-Convergence

In this part, we provide the theoretical proof of the feasibility of the influence function for deep networks (non-convergent) in [32]. In the derivation of the influence function, it's assumed that $\hat{\theta}$ could be the global minimum. However, if $\hat{\theta}$ is obtained in deep networks trained with SGD in a non-convex setting, it might be a local optimum and the exact influence can hardly be computed. Here we provide the proof in [32] on how can influence function approximate the parameter change in deep networks.

Consider a training point $z$. When the model parameters $\tilde{\theta}$ are close to but not at a local minimum, $I_{up,params}(z)$ is approximately equal to a constant (which does not depend on $z$) plus the change in parameters after upweighting $z$ and then taking a single Newton step from $\tilde{\theta}$. The high-level idea is that even though the gradient of the empirical risk at $\tilde{\theta}$ is not 0, the Newton step from $\tilde{\theta}$ can be decomposed into a component following the existing gradient (which does not depend on the choice of $z$) and a second component responding to the upweighted $z$ (which $I_{up,params}(z)$ tracks).

Let $g \stackrel{\text{def}}{=} \frac{1}{n}\sum_{i=1}^{n}\nabla_{\theta}L(z_i,\tilde{\theta})$ be the gradient of the empirical risk at $\tilde{\theta}$; since $\tilde{\theta}$ is not a local minimum, $g \neq 0$. After upweighting $z$ by $\epsilon$, the gradient at $\tilde{\theta}$ goes from $g \mapsto g + \epsilon\nabla_{\theta}L(z,\tilde{\theta})$, and the empirical Hessian goes from $H_{\tilde{\theta}} \mapsto H_{\tilde{\theta}} + \epsilon\nabla_{\theta}^2 L(z,\tilde{\theta})$. A Newton step from $\tilde{\theta}$ therefore changes the parameters by:

$$N_{\epsilon,z} \stackrel{\text{def}}{=} -\left[H_{\tilde{\theta}} + \epsilon\nabla_{\theta}^2 L(z,\tilde{\theta})\right]^{-1}\left[g + \epsilon\nabla_{\theta}L(z,\tilde{\theta})\right]. \tag{20}$$

Ignoring terms in $\epsilon g$, $\epsilon^2$, and higher, we get $N_{\epsilon,z} \approx -H_{\tilde{\theta}}^{-1}\left(g + \epsilon\nabla_{\theta}L(z,\tilde{\theta})\right)$. Therefore, the actual change due to a Newton step $N_{\epsilon,z}$ is equal to a constant $-H_{\tilde{\theta}}^{-1}g$ (that doesn't depend on $z$) plus $\epsilon$ times $I_{up,params}(z) = -H_{\tilde{\theta}}^{-1}\nabla_{\theta}L(z,\tilde{\theta})$ (which captures the contribution of $z$).

## B  Bias Removal via Machine Unlearning

### B.1  A Closer Look at the Toy Experiment

We conduct an experiment on a logistic regression task using Eq. 19. We simplify the Colored MNIST classification task to a binary classification problem of distinguishing between only digits 3 and 8, on a training set with a bias ratio of 0.95, 0.9 and 0.8, and a balanced test set. To be specific, a bias ratio of 0.95 means 95% bias-aligned samples <digit3_color3, digit8_color8> and 5% bias-conflicting samples <digit3_color8, digit8_color3> in the training set. We trained a regularized logistic regressor: $\text{argmin}_{w \in R^d}\sum_{i=1}^{n}l(w^T x_i, y_i) + \lambda\|w\|_2^2$. Fig. 5 (a) illustrates the classification results of the vanilla classifier (trained on the 0.95-biased train set) on part of test samples. We denote Digit by shape (triangle and rectangle) and Color by color (yellow and green). The solid line represents the learned classification boundary and the dotted line represents the expected classification boundary. It can

be observed that the learned classifier tends to classify digits according to their color. Based on the observed bias, we employ Eq. 19 to evaluate how each training sample contributes to the bias. In Fig. 5(b), we select and visualize the most helpful (reduce bias) and harmful (increase bias) samples. We found that the most harmful samples are bias-aligned while helpful samples are bias-conflicting. With this inspiration, We further visualize the influence distribution of training samples in Fig. 6. We denote the bias-conflicting sample with "red dot" and the bias-aligned sample with "blue dot". We find that most bias-aligned samples tend to be harmful while bias-conflicting samples tend to be helpful. This pattern is consistent across different ratios of bias-conflicting samples. Additionally, the influences of helpful samples are larger than those of harmful ones. Visualizations are produced with randomly 500 samples from the training set.

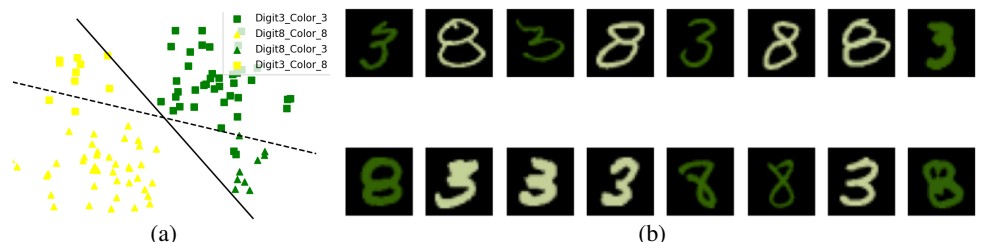

Figure 5: (a) Illustration of the learned pattern on our toy dataset. (b) Visualization of helpful samples (top row) and harmful samples (bottom row).

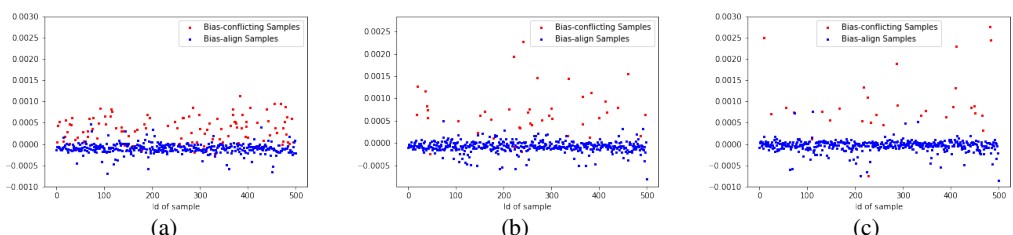

Figure 6: Influences of training samples with bias ratios of (a) 0.8, (b) 0.9, (c) 0.95.

Inspired by this observation, our unlearning strategy is further refined. Hence, we propose a straightforward solution that further mitigates the influence of a harmful sample with a bias-conflicting sample. Consequently, we update the parameters to unlearn the harmful samples by:

$$\theta_{new} = \hat{\theta} + \sum_{k=1}^{K} H_{\hat{\theta}}^{-1}(\nabla_{\hat{\theta}}L(z_k, \hat{\theta}) - \nabla_{\hat{\theta}}L(\bar{z}_k, \hat{\theta})), \tag{21}$$

where $\bar{z}_k$ denotes the bias-conflicting sample of $z_k$. Following the explanation in influence theory [32], our unlearn mechanism removes the effect of perturbing a training point $(\bar{a}, x, y)$ to $(a, x, y)$. In other words, we not only remove the influence caused by harmful sample $z_k$, but further ensure fairness with the corresponding counterfactual sample $\bar{z}_k$.

To further illustrate the functionality of Eq. 21, we measure the influences of the selected harmful and helpful sample pairs by:

$$I_{up,bias}(z_k, B(\hat{\theta})) = -\nabla_{\hat{\theta}}B(\hat{\theta})H_{\hat{\theta}}^{-1}(\nabla_{\hat{\theta}}L(z_k, \hat{\theta}) - \nabla_{\hat{\theta}}L(\bar{z}_k, \hat{\theta})), \tag{22}$$

with visualizations in Fig. 7. By calculating the difference between the harmful samples and helpful samples, the biased effect is significantly amplified. In this way, the unlearning becomes more effective.

## B.2 Deriving Alternative Efficient Unlearn

In the above sections, the unlearning process is based on the assumption that we could access the original training sample $z_k$ to identify and evaluate biases and then forget them. However, in practice,

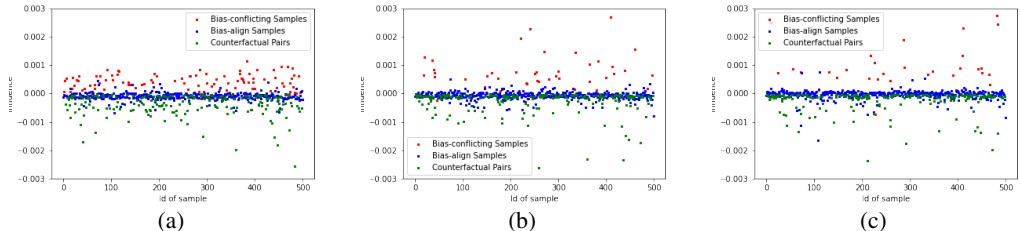

(a)                                    (b)                                    (c)

Figure 7: Influences of selected training sample (counterfactual) pairs in Eq. 21 with bias ratios of (a) 0.8, (b) 0.9, (c) 0.95.

the training set might be too large or even unavailable in the unlearning phase. In response, we further propose to approximate the unlearning mechanism with a small external dataset. As the influence to be removed can be obtained from the change of the protected attribute, we can construct the same modification to the protected attribute on external samples. In particular, we employ an external dataset $D_{ex}$ as in Section 3.1 in the main work to construct counterfactual pairs for unlearning, which redefines Eq. 21 as:

$$\theta_{new} = \hat{\theta} + \sum_i H_{\hat{\theta}}^{-1}(\nabla_{\hat{\theta}} L(c_i, \hat{\theta}) - \nabla_{\hat{\theta}} L(\bar{c}_i, \hat{\theta})). \tag{23}$$

As $D_{ex}$ can be easily obtained from an external dataset rather than the training set, e.g., the test set, the practical applicability of our method could be significantly enhanced.

We further visualize the influence of samples in the balanced external dataset in Fig. 8 (a). In the balanced dataset, the ratio of bias-aligned and bias-conflicting samples is about 50%. We can observe that the pattern of harmful bias-aligned samples and helpful bias-conflicting samples in the external dataset is similar to the training set. By comparing the influence of counterfactual pairs in the external dataset (Fig. 8 (b)) and the training set (Fig. 8 (c)), we can find the distributions are similar, which proves the feasibility of our alternative unlearning.

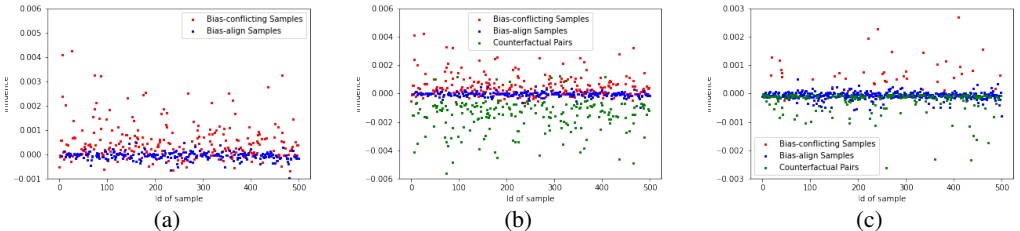

(a)                                    (b)                                    (c)

Figure 8: Influences of samples in (a) external dataset, (b) external dataset (with counterfactual sample pairs), (c) training set.

### B.3 Alternative Efficient Unlearn vs. Directly Unlearn Training Data.

Tackling the problem that, in practice, the training set might be too large or even unavailable in the unlearning phase, we propose an alternative unlearning strategy in Sec. 3.3 in the main work. We approximate the change of the protected attribute by constructing the same modification to the protected attribute on external samples. Then we unlearn the same perturbation from the model with the constructed external dataset. In Sec. 4.4 in the main work, we provide the performance comparison of alternative efficient unlearn (Ours) and directly unlearn training data (Eq. 7 and Eq. 8 in the main work).

In this section, we further compare the performance of alternative unlearning on Adult and simplified Colored MNIST on logistic regression, with results reported in Tab. 9 and Tab. 10. We can find that in five experiments, alternative learning achieves comparable performance with the two directly

unlearning strategies. Comparing Eq. 7 and Eq. 8, we can find that the modified Eq. 8 reaches convergence taking less iteration. The number of samples used is 200 for the two datasets.

| Attr. | Method | Bias ↓ | Time(s) | # Iter. | Acc.(%) ↑ |
|---|---|---|---|---|---|
| race | Vanilla | 0.0134 | - | - | 0.8259 |
| | Eq. 7 | 0.0002 | 1394 | 39 | 0.8249 |
| | Eq. 8 | 0.0002 | 1398 | 10 | 0.8311 |
| | Ours | 0.0002 | 0.0039 | 46 | 0.8229 |
| gender | Vanilla | 0.0494 | - | - | 0.8259 |
| | Eq. 7 | 0.0001 | 1386 | 212 | 0.8234 |
| | Eq. 8 | 0.0001 | 1390 | 186 | 0.8252 |
| | Ours | 0.0006 | 0.0038 | 252 | 0.8232 |

Table 9: Alternative Efficient Unlearn on Adult.

| Attr. | Method | Bias ↓ | Time(s) | # Iter. | Acc.(%) ↑ |
|---|---|---|---|---|---|
| 0.95 | Vanilla | 0.4624 | - | - | 0.5922 |
| | Eq. 7 | 0.1642 | 183 | 201 | 0.8548 |
| | Eq. 8 | 0.1624 | 183 | 157 | 0.8617 |
| | Ours | 0.1496 | 0.0017 | 74 | 0.8594 |
| 0.9 | Vanilla | 0.4086 | - | - | 0.6517 |
| | Eq. 7 | 0.1599 | 183 | 212 | 0.9102 |
| | Eq. 8 | 0.1562 | 183 | 185 | 0.9211 |
| | Ours | 0.1658 | 0.0018 | 77 | 0.9113 |
| 0.8 | Vanilla | 0.3735 | - | - | 0.6517 |
| | Eq. 7 | 0.1622 | 183 | 187 | 0.9241 |
| | Eq. 8 | 0.1617 | 183 | 169 | 0.9312 |
| | Ours | 0.1611 | 0.0017 | 67 | 0.9244 |

Table 10: Alternative Efficient Unlearn on Colored MNIST.

## B.4 Efficient Unlearning for Deep Networks.

In our experiment, We are inspired by Sec. 5.1 and Sec. 5.2 in [32] which keep all but the top layer in deep networks frozen and measure influence. We follow this setting so that the finetuning on deep networks can be simplified as logistic regression. In this part, we investigate the difference in finetuning different numbers of layers. The experiment is conducted on Colored MNIST with MLP with 3 hidden layers.

**Discussion on Different Fine-tuning Strategies.** Following Sec. 4.4 in the main work, we explore the impact of unlearning different numbers of layers (i.e., the top one, two, three MLP) on the Colored MNIST with three bias ratios, with results in Tab. 11. Interestingly, the accuracy excels with two layers but decreases with three layers. Additionally, fine-tuning multiple layers takes much longer time on computation on much more parameters. It is also worth noting that our method could achieve such superior or competing performance even only by updating the last layer in deep models, which calls for more in-depth analysis in the future.

## B.5 Effectiveness of Pre-calculating Hessian.

In Sec. 3.4 in the main work, we propose to pre-calculate the inverse Hessian before performing unlearning. In this way, we approximate the Hessian as it should change with model parameters, however, we prevent the large computation cost of updating and inverting the Hessian at every iteration. In this part, we empirically illustrate the effectiveness of our approximation. Experiments are conducted on Colored MNIST and Adult datasets with logistic regression tasks, with results provided in Tab. 12 and Tab. 13. "wo/" denotes unlearning without pre-calculation. It can be observed that unlearning with or without can achieve comparative performance on bias and accuracy. However, our method can save about 40% run time on Adult and 97% run time on Colored MNIST. The reason is that the number of parameters for Colored MNIST is much larger than Adult, so that the calculation of inverse Hessian makes up a larger proportion of the total run time.

| Ratio | Method | # Lay. | # Para. | Acc(%) ↑ | Bias ↓ | Time(s) |
|---|---|---|---|---|---|---|
| 0.995 | Vanilla | - | - | 38.59 | 0.5863 | - |
| | Ours[1] | 1 | 1000 | 62.34 | 0.3415 | 3.750 |
| | Ours[2] | 2 | 11000 | 64.18 | 0.3378 | 439.34 |
| | Ours[3] | 3 | 21000 | 55.32 | 0.3519 | 504.12 |
| 0.99 | Vanilla | - | - | 51.34 | 0.4931 | - |
| | Ours[1] | 1 | 1000 | 71.19 | 0.2757 | 3.750 |
| | Ours[2] | 2 | 11000 | 74.18 | 0.3134 | 432.86 |
| | Ours[3] | 3 | 21000 | 61.45 | 0.2949 | 496.44 |
| 0.95 | Vanilla | - | - | 77.63 | 0.2589 | - |
| | Ours[1] | 1 | 1000 | 86.39 | 0.1849 | 3.975 |
| | Ours[2] | 2 | 11000 | 87.34 | 0.1902 | 434.25 |
| | Ours[3] | 3 | 21000 | 86.47 | 0.1914 | 501.24 |

Table 11: Ablation on # MLP Layers.

| Attr. | Method | Bias ↓ | Time(s) | # Iter. | Acc.(%) ↑ |
|---|---|---|---|---|---|
| race | Vanilla | 0.0134 | - | - | 0.8259 |
| | wo/ | 0.0002 | 0.0064 | 42 | 0.8229 |
| | Ours | 0.0002 | 0.0039 | 46 | 0.8229 |
| gender | Vanilla | 0.0494 | - | - | 0.8259 |
| | wo/ | 0.0006 | 0.0066 | 149 | 0.8243 |
| | Ours | 0.0006 | 0.0038 | 252 | 0.8232 |

Table 12: Efficient Hessian Computation on Adult.

# C  Experiment Details

## C.1  Dataset

**Colored MNIST.** Colored MNIST is constructed based on the MNIST dataset [68] designed for digit classification tasks. To build a biased correlation, ten distinct RGB values are applied on grayscale digit images [3, 69, 70]. Digit and color distribution are paired to build biased correlations in the training set. Bias-aligned samples are defined as fixed combinations of digit and color like Digit 1, Color 1 while bias-conflict samples are defined as other combinations like Digit 1, random Color in 2-10. In our Experiment, we use 3 different training sets by setting different bias ratios 0.995, 0.99, 0.95 for biased-aligned training samples where the ratio represents the partition of bias-aligned samples in the training set. The higher the ratio, the higher the degree of bias. The split of the training set, test set, and external set is 60000, 10000, and 10000.

**CelebA.** CelebA dataset [71] is a face recognition with 40 types of attributes like gender, age (young or not), and lots of facial characteristics (such as hair color, smile, beard). The dataset contains a total of 202,599 images which, following the official train validation split, consists of 162,770 images for training and 9,867 images for testing. We choose Gender as the protected attribute, Hair-color (blonde hair or not) and Attractive as the target attribute following [7, 8]. The number of selected samples for the two target attributes is 200 and 182, which are split from the test set.

| Ratio | Method | Bias ↓ | Time(s) | # Iter. | Acc.(%) ↑ |
|---|---|---|---|---|---|
| 0.95 | Vanilla | 0.4624 | - | - | 0.5922 |
| | wo/ | 0.1490 | 0.0556 | 59 | 0.8674 |
| | Ours | 0.1496 | 0.0017 | 74 | 0.8594 |
| 0.9 | Vanilla | 0.6517 | - | - | 0.4086 |
| | wo/ | 0.1698 | 0.0498 | 46 | 0.9093 |
| | Ours | 0.1658 | 0.0018 | 77 | 0.9113 |
| 0.8 | Vanilla | 0.2857 | - | - | 0.6915 |
| | wo/ | 0.1689 | 0.0517 | 34 | 0.9264 |
| | Ours | 0.1611 | 0.0017 | 67 | 0.9244 |

Table 13: Efficient Hessian Computation on Colored MNIST.

**Adult Income Dataset.** The Adult dataset is a publicly available dataset in the UCl repository [72] based on 1994 U.S. census data. The goal of this dataset is to successfully predict whether an individual earns more or less than $50,000 per year based on features such as occupation, marital status, and education. We follow the processing procedures in [41]. In our experiment, we choose gender and race as protected attributes following [73, 74]. We split 200 samples from the test set as the external dataset.

## C.2    Baselines

For the sanity check experiment on a toy Colored MNIST dataset, we use a vanilla logistic regression model as the baseline. For experiments with deep networks, we compare our method with one pre-processing baseline Reweigh [77], 6 in-processing debiasing baselines (LDR [25], LfF [78], Rebias [79], DRO [7], SenSEI [80], and SenSR [81]) and 4 post-processing baselines (EqOdd [35], CEqOdd [35], Reject [82] and PP-IF [83]). [77] utilizes the influence function to reweight the training sample, in order to re-train a fair model targeting group fairness metrics (equal opportunity and demographic parity). Among in-processing baselines, LDR, LfF, Rebias, and DRO are designed explicitly to target higher accuracy (on unbiased test set or worst-group test set) and implicitly target fairness, while SenSEI and SenSR are designed to target individual fairness. EqOdd, CEqOdd and Reject are designed to target different group fairness metrics (equal odd and demographic parity), while [83] proposes a post-processing algorithm for individual fairness.

**Baselines for experiment on Large Language Model.** We evaluate several baseline debiasing methods. **Counterfactual Data Augmentation (CDA)** [94] adjusts a corpus for balance by exchanging words indicative of bias (such as 'he/she') within the dataset. This newly balanced corpus is then typically utilized for additional model training to reduce bias. **Dropout** [95] suggests enhancing dropout rates and incorporating an extra pre-training stage for debiasing. **SentenceDebias** [88] aims to derive unbiased representations by removing biased projections on a presumed bias subspace from the original sentence representations. **Iterative Nullspace Projection (INLP)** [86] employs a projection-based approach to exclude protected attributes from representations. Finally, **Self-Debias** [87] advocates for the use of a model's intrinsic knowledge to avert the generation of biased text.

## D    Discussion

### D.1    Dataset Generation

In our experiments, we utilize approximated counterfactual samples for CelebA due to the unavailability of strict counterfactual data. Based on attribute annotations, we select images with the same target attributes but opposite sensitive attributes, while maintaining other attributes as much as possible. Our method achieves the best results on the worst-case group, indicating that the approximated counterfactual samples can also effectively enhance fairness in predictions. Similar to our approach, [96] proposes to select pairs of counterfactual images based on attribute annotations on the CUB dataset to produce counterfactual visual explanations. Their experiments also show that neural networks can discern major differences (such as gender in our work) between images without strict control (such as background).

For real-world visual datasets (like facial dataset or ImageNet), the unavailability of strict counterfactual data is a common challenge. Existing methods propose to train a generative model to create counterfactual images with altered sensitive attributes [97–99], which seems to be a viable approach for obtaining counterfactual datasets for more diverse vision applications. Building upon these methods, we will extend our approach to more scenarios.

### D.2    Influence Estimation

In our unlearning experiment, we freeze the parameters of all other layers except the top layer. Previous work investigates the estimation accuracy of the influence function on both multi-layer and single-layer setups [100]. It performs a case study on the MNIST. For each test point, they select 100 training samples and compute the ground-truth influence by model re-training. Results show that estimations are more accurate for shallow networks.

Our results in Tab. 7 in the main manuscript also validate this point. When applying FMD to a three-layer neural net, the performance on either accuracy or bias becomes worse. This could potentially be attributed to the inaccurate estimation of influence function on multi-layer neural nets. In our experiments, we adhere to the set-up in [32], where the influence function is only applied to the last layer of deep models, which proves to be effective.

As verified in [32, 100, 101], influence estimation matches closely to leave-one-out retraining for logistic regression model. As discussed in [97], measuring influence score for the last layer can be regarded as calculating influence from a logistic regression model on the bottleneck features (Sec. 5.1 in the main manuscript). The same setup is followed by many influence function-based works [102, 103] and proves to be effective.

### D.3 Computational Complexity

As for bias-effect evaluation, with $n$ training points and $\theta \in R^d$, directly computing Eq. 5 (in the main manuscript) requires $O(nd^2 + nd^3)$ operations. In our experiment, we only activate the last layer so that d is small. However, when the number of training samples is very large, performing Eq. 5 is expensive. As for the debiasing phase, it requires $O(nd^2 + kd^2)$ operations, where k is the number of samples to unlearn. Note that if hessian is calculated in the bias-effect evaluation phase, it can be directly used in the debiasing phase. Hence, the overall computational complexity using Eq. 7 and Eq. 8 is $O(nd^2 + kd^2 + nd^3)$.

However, in our proposed alternative debiasing method, we only utilize an external counterfactual dataset with a small number of k. Hence, we can omit the $O(nd^3)$ operations to compute influences and rank the training samples. Hence, the overall computational complexity using Eq. 9 (Ours) is $O(nd^2 + kd^2)$. Experimental comparison results can be referred to Tab. 5 (in the main manuscript). Debiasing with Eq. 8 takes about 500x more time than Eq. 9 (in the main manuscript).

## E  Preliminaries

### E.1  Influence Function

The origins of influence-based diagnostics can be traced back to important research papers such as [64, 104, 105]. More recently, Koh and Liang [32] introduced the concept of influence functions to large-scale deep learning, which numerous publications have since followed up. In their work, [32] advocated for the use of an approximation, Eq. 13, to estimate the change in loss when a small adjustment is made to the weights of the dataset. In practical applications involving deep models, the Hessian matrix (H) cannot be stored in memory or inverted using standard linear algebra techniques. However, by considering a fixed vector (v), the Hessian vector product (HVP), Hv, can be computed in O(bp) time and memory [106], where b represents the batch size and determines the number of training examples used to approximate H (for a given loss function L). The iterative procedure LISSA [107], employed by [32], relies on repeated calls to the HVP to estimate the inverse HVP.

### E.2  Counterfactual Fairness

Counterfactual fairness, a relatively new concept, has emerged as a means to measure fairness at an individual level [31]. The fundamental idea behind this approach is to determine the fairness of a decision for an individual by comparing it with the decision that would have been made in an alternate scenario where the individual's sensitive attributes possessed different values. This concept builds upon earlier work [108], which introduced a causal framework for learning from biased data by examining the relationship between sensitive features and the data. Recent advancements in deep learning have further contributed to this field, with novel approaches [46, 45, 109, 98] proposing methods to enhance the accuracy of decision-making models by improving the approximation of causal inference, particularly when dealing with unobserved confounding variables.

### E.3  Demographic Parity and Equal Opportunity

**Demographic Parity** [34]: A predictor $Y$ satisfies demographic parity if $P(Y|A = 0) = P(Y|A = 1)$, where $A$ is the sensitive attribute. The likelihood of a positive outcome should be the same regardless of whether the person is in the protected (e.g., female) group.

**Equal Opportunity** [35]: "A binary predictor $Y$ satisfies equal opportunity with respect to $A$ and $Y$ if $P(Y = 1|A = 0, Y = 1) = P(Y = 1|A = 1, Y = 1)$ " . This means that the probability of a person in a positive class being assigned to a positive outcome should be equal for both protected and unprotected (female and male) group members.

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
