# OpenReview forum: "Fast Model DeBias with Machine Unlearning"
_NeurIPS.cc/2023/Conference — NeurIPS 2023 poster_

### Official Review · Reviewer_hkdb · 2023-07-01

**Soundness:** 2 fair
**Presentation:** 3 good
**Contribution:** 2 fair
**Rating:** 5
**Confidence:** 4

**Summary:**

This paper proposes an approach to removing biases from trained models without necessitating full retraining and without necessitating access to a large dataset that has annotations w.r.t biases. The proposed method consists of 3 steps: firstly, identifying which attributes are ‘biased’ (according to a counterfactual definition of bias: the more the predictions differ between the factual and counterfactual examples, the more biased the model is). Second, identify which examples are those contributing to biased behavior, using influence functions. Finally, once the (top K) harmful samples are identified, they run an unlearning procedure (via a Newton step) to remove the influence of those examples.


**Strengths:**

- This paper tackles an important problem and provides useful motivation and intuition for understanding the issues with biases in neural networks

- The proposed approach makes fewer limiting assumptions than some of the previous methods, and the authors show that empirically, it outperforms certain baselines on certain metrics on some (simple) datasets / models

- The paper is well-structured and mostly clear, with some exceptions (see weaknesses section below)


**Weaknesses:**

A) Clarity can be significantly improved in some sections. Some particular examples:

a) Some imprecise statements, e.g. what does it mean to be “closely tied” (line 136)

b) In line 158, what is z_k? Should that have been z_i? (i is the variable that the sum is over, but it’s not used, unless I’m missing it?). Also, need to define n in that equation.

c) In Equation 3 (which defines the counterfactual bias), how do we choose the value to modify a_i with? Does the equation need some expectation over which alteration to make to the affected attribute?

d) in the “Biased-effect evaluation section”, the authors say that they investigate how the bias is learned. More precisely, though, what they do is estimate the influence of specific training examples, so a more appropriate description is to identify which examples lead to learning the bias, not *how* the bias is learned.

e) What does “up” mean / stand for, in e.g. Equation 6.

f) Generally, the Bias-effect evaluation paragraph is hard to understand without a background in influence functions. It would be great to give more intuition around equations.

g) In the experiments section, the authors don’t describe the baselines they use, making it harder for someone unfamiliar with this literature to assess their appropriateness and the value of the associated comparisons


B) [Terminology] Unlearning refers to removing the influence of a subset of training examples from a trained model. Therefore, doing “unlearning” on external samples (e.g. from a test set) doesn’t make sense according to that definition. Would be useful to clarify this point and perhaps use a different term instead. Debiasing sounds like a more appropriate word to me rather than unlearning (though inspiration may be drawn from unlearning methods).

C) Issues with the experimental setup:

a) there are no confidence intervals in the tables, making it hard to assess whether the proposed method’s improvement over the considered baselines in each case is statistically significant. Sometimes the results are quite similar.

b) In Table 1, why is DRO excluded from comparisons? In Table 3, why don’t the authors also show the Unb. and Wor. columns, as in Table 2? Those seem like a more direct and appropriate way to assess the degree of biasing in a way that actually matters / affects the model’s behavior, rather than only looking at the Bias column.

c) Overall, the experiments are conducted on simple datasets with relatively small neural networks, and it’s unclear how this affects the conclusions. For instance, it’s surprising that unlearning only the top layer is successful, as in general this is not a good unlearning method (examples typically influence the trained model throughout layers, in a highly non-convex manner)

d) It would also be nice to compare against the oracle for unlearning, to get an upper bound. Specifically, once the harmful examples are identified, the oracle for unlearning would simply remove them from the training set and retrain the model without them (this of course only applies to the case where unlearning is done using training examples, as is the setup in unlearning). How well does the proposed approach debias, compared to this oracle for the unlearning step?

e) Looking at Table 5, counterintuitively, Eq. 9 (“unlearning” using external samples) leads to smaller bias than Eq. 7 (unlearning using training samples). This seems to contradict the narrative that the former should do worse than the latter but is adopted instead due to not wanting to assume access to the training set? Do the authors have an explanation for this? Or is it the case that all these results are statistically indistinguishable from each other? (again, there are no confidence intervals in the tables, making it hard to tell.)


**Questions:**

- How did the authors pick the value of K (number of harmful samples to unlearn)? Is this different for each dataset / setting?

- When computing the time (for quantifying the efficiency of the method), which steps are taken into account for the proposed method? Are all steps 1-3 counted towards this? If not, why not? Is this done fairly w.r.t how it’s counted for other methods?

- Follow-up: what is the computational complexity of running the proposed method, in all 3 steps? Computing Hessians is expensive, and it sounds like the proposed method is to compute the influence of each example, in order to then rank them and find the top K.

- How well does the method scale to deeper neural networks? All architectures used in the experiments are very small.

- When using an external dataset for “unlearning”, how similar does that dataset need to be to the training set, in order for that to be beneficial? What if the external dataset is distributionally-shifted in different ways?

- Could some of the baselines considered also be (easily) modified to use external datasets and / or fewer samples?


**Limitations:**

- The proposed method requires datasets with access to attributes, so that counterfactuals can be created by e.g. changing the value of one attribute while keeping the values of other attributes fixed. But in practice, we typically don’t have attribute information for datasets of interest.

---

> ### Author Rebuttal · Authors · 2023-08-09
>
> We appreciate the reviewer for the detailed comments! Please find our replies to your questions below (**Q denotes Question, W denotes Weakness and R denotes Reviewer**). **Due to the space limitation, answers to Weakness continues in the global response.** Sorry for the inconvenience. **Please note that results for WC-abd and Q2 are included in Tab.1 in the PDF document.**
>
> >Q1. *value of K* :
>
> See response to Reviewer NRrd Weakness 3.
>
> >Q2. *time counting*:
>
> The reported time of our method is counted on step 3 (debiasing phase). For fair comparison, we also reported the debiasing time of the baseline methods. As an additional reference, we included the total process time (in Tab.1 in the PDF document). The results demonstrate that our method remains efficient in terms of time.
>
> >Q3. *computational complexity*:
>
> Thanks for the comments. As for bias-effect evaluation, with $n$ training points and $\theta \in R^d$, directly computing Eq. 5 requires $O(nd^2 + nd^3)$ operations. In our experiment, we only activate the last layer so that d is small. However, when the number of training samples is very large, performing Eq. 5 is expensive. As for the debiasing phase, it requires $O(nd^2 + kd^2)$ operations, where k is the number of samples to unlearn. Note that if hessian is calculated in the bias-effect evaluation phase, it can be directly used in the debiasing phase. Hence, the overall computational complexity using Eq. 7 and Eq. 8 is $O(nd^2 + kd^2 + nd^3)$.
>
> However, in our proposed alternative debiasing method, we only utilize an external counterfactual dataset with a small number of k. Hence, we can omit the $O(nd^3)$ operations to compute influences and rank the training samples. **Hence, the overall computational complexity using Eq. 9 (Ours) is** $O(nd^2 + kd^2)$.
>
> Experimental comparison results can be referred to Table 5. Debiasing with Eq. 8 takes about 500x more time than Eq. 9.
>
> >Q4. *large model*:
>
> See response to Reviewer 2Mrc Weakness 2.
>
> >Q5. *distribution shift*:
>
> In our experiment, we randomly split the external dataset from the whole dataset so that it is i.i.d with the training set.
>
> We further experiment on unbalanced external datasets (**long-tailed MNIST**) with imbalanced factors of 5, 10, and 20, **i.e., distribution-shift**. The results demonstrated that an **imbalanced external dataset can lead to a decline in performance**. Thank you for providing the inspiration, and we agree that it is an interesting topic. How to prevent the influence of distribution shifts is not included in our problem setting, however, will be discussed in our future works.
>
> Worth mentioning is that in our experiments on large models (tested on StereoSet), we also use a **distributionally-shifted dataset (Crows-Pairs)** for debiasing. The debiasing results are promising, but it is possible that some performance limitations exist due to the differences in distributions between datasets.
>
> | Bias Ratio | Method | Imbalanced Factor | Acc.(%) ↑ | Unb. | Wor. | Bias ↓ |
> | --- | ---|---|--- | --- | --- | --- |
> | 0.99 | Vanilla | - | 51.34±3.37 | 73.06±3.17 | 46.41±7.28 | 0.4931 |
> |  | LDR | - | 76.48±1.45 | 87.18±1.43 | 74.93±3.49 | 0.2511 |
> |  | LfF | - | 64.71±2.17 | 80.42±1.94 | 61.83±4.29 | 0.2366 |
> |  | Rebias | - | 80.41±1.03 | 87.85±0.95 | 76.19±1.98 | 0.2302 |
> |  | DRO | - | 71.33±1.57 | 83.82±0.87 | 68.24±2.27 | 0.2563 |
> |  | Ours | **1** | 80.04±1.42 | 87.69±1.03 | 76.31±2.03 | 0.2042 |
> |  | Ours | **5** | 78.28±1.37 | 83.08±1.31 | 67.07±2.44 | 0.235 |
> |  | Ours | **10** |76.90±1.64 | 82.33±1.49 | 65.54±2.41 | 0.2514 |
> |  | Ours | **20** | 76.16±1.61 | 80.84±1.27 | 62.48±2.17 | 0.2581 |
> |  | Oracle | - | 82.74±1.09 | 89.08±0.98 | 79.03±1.84 | 0.1683 |
>
> >Q6. *baseline*:
>
> For additional reference, we modify our baselines to be trained on the same number of samples as our method (Experiment conducted on Colored MNIST). The results indicate that reducing the data volume consistently leads to a decline in the performance of the baseline method.
>
> | Bias Ratio | Method | Acc.(%) ↑ | Bias ↓ | Time(s) | # Samp. |
> |---|---|---|---| --- | --- |
> | 0.99 | Vanilla| 34.62±3.17 | 0.5731 | - | - |
> |  |LDR| 76.48±1.45 | 0.2511 | 1,330 | 50000 |
> |  |LfF|64.71±2.17 | 0.2366 | 726 | 50000 |
> |  |Rebias| 80.41±1.03 | 0.2302 | 1,658 | 50000 |
> |  |DRO|71.33±1.57 | 0.2563 | 4,362 | 50000 |
> |  |LDR| 65.62±1.61 | 0.3123 | 317 | **5000**|
> |  |LfF |59.38±2.11 | 0.2983 | 196 | **5000** |
> |  |Rebias | 73.24±1.47 | 0.2581 | 408 | **5000** |
> |  |DRO | 65.24±3.28 | 0.2674 | 537 | **5000** |
> |  |Ours | 80.04±1.42 | 0.2042 | 59 (+33) | 5000 |
>
> >Q7. *dataset generating*:
>
> See response to **Reviewer AWfs Weakness2**.
>
> > WA-a: *clarity*
>
> In line 136, we intended to convey "highly correlated" as "closely tied". We will clarify it in the revision.
>
> > WA-b: *clarity*
>
> It should indeed be z_i in line 158. n is the number of training samples. We will further proofread our manuscript.
>
> > WA-c: *value of a_i*
>
> According to the definition of counterfactual fairness, the prediction should not change for all attainable a_i. **We randomly choose a_i from all attainable values (the set of values for a_i in the training set).** For example, in our experiment on Colored MNIST, with ten distinct colors included in the dataset, the alteration set is denoted as {color1, color2, ..., color10}. When modifying $a_i$, we assign a random color from the alteration set to $a_i$. The equation needs expectation over alterations.
>
> > WA-d: *clarity*
>
> In this section, our goal is to quantify how each training point $z$ in the training set $D_{tr}$ contributes to $B$. We agree with the reviewer that it should be “*which examples lead to learning the bias*”. We will clarify it in the revision.
>
> > WA-e: *"up"*
>
> The word "up" is the abbreviation of "upweight". Here we follow the subscript "up" used in [1]. We will further clarify this equation in our manuscript.
>
>  **Due to the space limitation, answers to Weakness continues in the global response.**

---

> > ### Author Response · Authors · 2023-08-17
> >
> > Dear Reviewer hkdb,
> >
> > We greatly appreciate your valuable time and thoughtful feedback. We have thoroughly revised our paper and responded in accordance with your suggestions, aiming to address all your concerns. We would be grateful if you would kindly let us know of any other concerns and if we could further assist in clarifying any other issues.
> >
> > Thank you once again for your contributions, and we extend our best wishes.
> >
> > Sincerely,
> >
> > Authors

---

> > ### Comment · Reviewer_hkdb · 2023-08-17
> > **thank you for the responses**
> >
> > Hi authors,
> >
> > Thank you for the thorough responses in your rebuttal.
> > The rebuttal clarifies some issues, e.g. about terminology, additional info about baselines, computational complexity. The authors also ran a number of additional experiments using additional baselines, larger models, imbalanced and distributionally-shifted datasets which strengthen the paper in my view.
> > While some weaknesses remain (notably requiring access to datasets with attributes; which sounds like can be mitigated to some extent but is still an open research question), I believe this paper's contribution is valuable, and I'm increasing my score to reflect that.

---

> > > ### Author Response · Authors · 2023-08-18
> > >
> > > Dear Reviewer hkdb,
> > >
> > > We extend our sincere gratitude for your thorough and insightful review of our manuscript, particularly in regards to aspects such as problem setting, writing clarity, and experimental analysis. Your feedback has been immensely valuable.
> > > We intend to incorporate the additional experiments and discussions that you have suggested into our revised manuscript. Furthermore, we are fully dedicated to enhancing the quality of our writing as per your recommendations.
> > > Your constructive feedback has been instrumental in shaping the improvement of our work. We greatly appreciate your acknowledgment of our response and the conducted experiments in our rebuttal.
> > >
> > > Thank you once again for your valuable comments. We hold high regard for your efforts in reviewing our manuscript. Wishing you all the best.
> > >
> > > Warm regards,
> > >
> > > The Authors

---

### Official Review · Reviewer_hEy4 · 2023-07-04

**Soundness:** 3 good
**Presentation:** 2 fair
**Contribution:** 3 good
**Rating:** 6
**Confidence:** 4

**Summary:**

The paper addresses the issue of biases in trained models, which are often caused by imbalances in the dataset used for training. The conventional approach to mitigating biases involves finetuning the model, but this incurs additional training costs. The paper proposes an alternative method called Fast Model Debiasing (FMD) that does not require finetuning.

The FMD method consists of three steps: identifying the bias attributes, identifying the top samples that contribute to the bias, and removing the bias. The key contribution of the paper is the use of Influence Functions (IF) and its extension to identify and eliminate biases. This approach is interesting and appears to be reasonable.



**Strengths:**

The main contribution of the proposed method is to leverage the IF (Implicit Fairness) technique and its extension to identify and remove bias in models. This approach offers a practical solution for making models debiased.

Compared to typical fine-tuning solutions, this method avoids the need for additional training effort, making it a more efficient procedure. By directly removing bias through the IF technique, the model can be made fairer without requiring extensive retraining.

Furthermore, this method can be implemented as a separate API, allowing for easy integration into existing systems. This API can be used to remove bias from models, ensuring fairness in various applications.

**Weaknesses:**

Here, the general idea is simple: to identify the samples, and further enhance the model to debias.
More experiments should be added to verify this.
refer to the following: limitation parts.

**Questions:**

In the approach, several layers in the model are updated based on certain rules to remove the bias. To further extend this work, it would be interesting to explore which modules are sensitive to bias and contribute to the debiasing process.
For example, when considering model adaptation during the inference stage, we can optimize the Batch Normalization (BN) module by re-estimating the mean and variance based on the fresh samples from new scenarios in just a few steps. HEre, BN is the sensitive for model adaptation.
Back to the debias, which module is the key?

There are some other questions:
1. Select K top harmful samples. what is the suggested value for K?
2. Such harmful samples highly depends on the testing samples. what is such testing samples from?
-if testing samples is from one subset of the training/validation set, how does the proposed not involve new bias after removing the influence of K top harmful samples?
3. Not find the bassline accuracy of the model without the debias, and how to guarantee that the updated parameters would not impact the model accuracy?
4. Have you tried other complex or big model?



**Limitations:**

1. In the proposed FMD,  based on the Newton update, the model is updated. The model accuracy generally would be impacted. But, most of the given experiments show that both of accuracy and debias performance are best. Such results might not be convinced. All of the experiments have been conducted using small or simple models on easy tasks. It is important to verify whether such methods can be applied to other typical or complicated tasks/datasets, such as ImageNet/COCO or segmentation tasks. Please enable more experiments or cite more other reference.
* Firstly, add one line or column to show the baseline performance, e.g., the accuracy without the debias.
* Secondly, extend to more scenarios, e.g., imageNet dataset.
* Thirdly, maybe one flow figure is helpful to understand the overall FMD procedure.
* Model debias is partially related to other topics, e.g., OOD, model, domain adaptation or others, where some debias method are also introduced. if time and space are ok, survey and add more reference.

2. Implementing such a scheme requires building a specific dataset (even if it is small) and updating the model layers, which can be a time-consuming process. It might not be easier than performing a quick finetuning for debiasing. Therefore, it would be beneficial to compare the complexity between the proposed approach and quick finetuning, as well as evaluate their performance.
* In general, quick finetuning does not necessarily require a larger dataset. A small specific dataset with data augmentation might be sufficient for finetuning to address bias.

---

> ### Author Rebuttal · Authors · 2023-08-09
>
> Thanks for the constructive comment！
>
> >Q1. *which module is the key:*
>
> In our method, we leverage the influence functions [1] to identify training samples that lead to biases,  which are not originally designated to identify specific machine learning modules that lead to bias. The debiasing process in our FMD is through the updating of the top 1-3 MLP layers of the biased model with the counterfactually constructed external datasets. Instead of telling which module is the key, our method mainly focus on identifying which data samples are the key (i.e., leading to bias or can be used for debiasing) and then update the top layers accordingly.
>
> Following your suggestion and to our knowledge, there is another line of work in the field of model editing [2, 3] which could be helpful to identify the key modules for debiasing. For instance, the Locate-Then-Edit [4, 5] method initially identifies parameters corresponding to specific knowledge and then modifies them through direct updates to the target parameters.
>
> Again many thanks for this valuable comment, which present a new promising direction for model debiasing. In our future work with large models (LMs), we will delve deeper into investigating the impact of different modules within LMs for fairness.
>
> >Q2. *value for K:*
>
> In our experiment, we select the number of samples **k=5000** for Colored MNIST, and **k=200** for both Adult and CelebA.
>
> We present experiments of debiasing using Eq. 9 with different values of K (on Colored MNIST with a bias ratio of 0.99) as an example. It can be observed that accuracy and bias improve gradually and tend to be saturated as the number of samples increases, while the time for unlearning (debiasing) continually increases. Hence, in experiments, we select the number of samples based on both the performance and time cost.
>
> | Method | # of samples (K) | Acc.(%) ↑ | Bias ↓ | Time(s) |
> | --- | --- | --- | --- | --- |
> | Vanilla | N/A | 51.34 | 0.4931 | - |
> | Ours | 500 | 73.67 | 0.2757 | 3.750s |
> |  | 1000 | 75.70 | 0.2648 | 7.396s |
> |  | 2000 | 77.36 | 0.2451 | 14.60s |
> |  | 3000 | 78.98 | 0.2232 | 25.37s |
> |  | 5000 | 80.04 | 0.2042 | 48.37s |
> |  | 6000 | 80.06 | 0.2044 | 58.24s |
> |  | 8000 | 80.31 | 0.2044 | 84.90s |
> |  | 10000 | 80.42 | 0.2035 | 105.25s |
>
> >Q3. *testing samples:*
>
> In our experiments with external datasets, the testing samples (the samples for debiasing) **have no overlap** with the train/val datasets. We agree with the reviewer that the debiasing process which might inadvertently introduce new biases. While based on our problem setting, these “potential” new biases would not affect our main objective to remove some specific bias . Some recent research also study the compatibility of different fairness criteria [6, 7, 8], and we will include the discussion of this issue in the revised version.
>
> >Q4. *baseline accuracy:*
>
> Sorry for the confusion, we use **"Vanilla"** to denote the model without any debiasing method, which is supervised trained with cross-entropy loss. For example, in our experiment on Adult (in Tab. 3), where training and test data are i.i.d., Vanilla achieves the best accuracy cause it fits the training set quite well. Compared to baseline methods, our proposed method achieved the best fairness score with second-best accuracies (just lower than Vanilla), indicating that our method ensures stable accuracy by only requiring rapid updates of a small set of parameters. In contrast, other methods sacrifice more accuracy to achieve certain levels of fairness.
>
> >Q5. *big model:*
>
> See Response to Reviewer 2Mrc Weakness 2.
>
> >Limitation. 1. *flow figure & reference:*
>
> We have included a flow figure in the PDF document. Model debiasing has close connections with out-of-distribution (OOD) detection and domain adaptation [9, 10], and we will discuss these relationships in our paper.
>
> >Limitation. 2. *time-consuming & quick finetuning:*
>
> **Complexity:** In our experiments, as for **Colored MNIST**, it takes about **4.3s** to generate the whole external dataset (5000 samples) on a single AMD CPU. For **Adult**, flipping sensitive attributes for the external dataset takes about **0.51s (200 samples)**. For **CelebA**, constructing counterfactual pairs takes about **11.2s (200 samples)**.
>
> **Experiment on finetuning:** We conducted experiments on Colored MNIST by fine-tuning the model on an **unbiased** dataset (i.e. bias ratio = 0) with 5000 (and 500) samples. Results are reported in the following table. It can be observed that our method outperforms Fine-tuning on bias ratios of 0.95 and 0.99.
>
> | Bias Ratio  | Method | Acc.(%) ↑ | Bias ↓ | Time(s) | # Samp. |
> | --- | --- | --- | --- | --- | --- |
> | 0.95 | Vanilla | 77.63 | 0.2589 | - | -|
> |  | LDR | 90.42 | 0.2334 | 19min40s | 60000 |
> |  | LfF | 85.55 | 0.1264 | 12min04s | 60000 |
> |  | Rebias | 89.63 | 0.1205 | 28min34s | 60000 |
> |  | Ours | 89.26 | 0.1189 | 56.44s | 5000 |
> |  | Fine-Tuning | 83.53 | 0.1656 | 199.57s | 5000 |
> |  | Fine-Tuning | 79.02 | 0.2176 | 116.17s | 500 |
> | 0.99 | Vanilla | 51.34 | 0.4931 |-|-|
> |  | LDR | 76.48 | 0.2511 | 22min10s | 60000 |
> |  | LfF | 64.71 | 0.2366 | 12min06s | 60000 |
> |  | Rebias | 80.41 | 0.2302 | 27min38s | 60000 |
> |  | Ours | 80.04 | 0.2042 | 48.37s | 5000 |
> |  | Fine-Tuning | 79.12 | 0.1971 | 198.37s | 5000 |
> |  | Fine-Tuning | 75.83 | 0.2105 | 118.03s | 500 |
>
>
>
> [1] Understanding black-box predictions via influence functions.
>
> [2] Fast model editing at scale.
>
> [3] Memory-based model editing at scale.
>
> [4] Knowledge neurons in pretrained transformers.
>
> [5] Locating and editing factual associations in GPT.
>
> [6] On the apparent conflict between individual and group fairness.
>
> [7] Fairness in criminal justice risk assessments: The state of the art.
>
> [8] Counterfactual fairness.
>
> [9] Reducing gender bias in neural machine translation as a domain adaptation problem.
>
> [10] Graphde: A generative framework for debiased learning and out-of-distribution detection on graphs.

---

> > ### Comment · Reviewer_hEy4 · 2023-08-16
> >
> > Thanks for your response, the rebuttal mostly addresses my concerns. the additional experiments results are valuable. I support such work can be accepted.

---

> > > ### Author Response · Authors · 2023-08-16
> > > **Replying to Reviewer hEy4**
> > >
> > > Dear Reviewer hEy4,
> > >
> > > We are very delighted with your support! We appreciate your valuable comments on our paper. We are committed to incorporating the additional experiments (extension to large models & ablation studies) and clarifications (flow figure & related works) into our revised manuscript.
> > >
> > > Once again, thank you very much! Best wishes to you!
> > >
> > > Warm regards,
> > >
> > > The Authors

---

### Official Review · Reviewer_NRrd · 2023-07-05

**Soundness:** 3 good
**Presentation:** 2 fair
**Contribution:** 3 good
**Rating:** 7
**Confidence:** 3

**Summary:**

In this paper, the authors propose Fast Model Debiasing (FMD), which targets solving biased prediction problems. The proposed FMD can effectively and efficiently remove model bias by leveraging only a small external dataset by altering the attribute, then evaluating the biased-effect of each sample, and finally using the most harmful samples to unlearn the model bias with the Newton update step. Experiments on Colored MNIST, CelebA, and Adult datasets demonstrate the effectiveness of the proposed method.

**Strengths:**

The motivation and the idea proposed in this paper are reasonable. The unlearning progress is efficient since the model parameters only need to be updated in several steps. The execution time is greatly shorter than the compared methods. The experimental results are sufficient to support the effectiveness of the proposed method.

**Weaknesses:**

1. Some writing issues. "Fast Model Debias" in the title, but "fast model debiasing" in the abstract. The full name of "FMD" is not explained when it first appeared in Sec 1.
2. In Eq 3, reduant ")".
3. Some usage of notation is not standard. For example, the function "B" accepts three kinds of input.
4. In line 231, the notation $C_i$ should be $c_i$.
5. The number of K when selecting harmful samples, and the bias threshold seems not explained in the experiments.

**Questions:**

N/A

---

> ### Author Rebuttal · Authors · 2023-08-08
>
> > Weakness. 1. *Some writing issues. "Fast Model Debias" in the title, but "fast model debiasing" in the abstract. The full name of "FMD" is not explained when it first appeared in Sec 1.*
> >
>
> We are really sorry for the mistake. The title should be "Fast model debiasing with machine unlearning". We will modify the title.
>
> > Weakness. 2. *In Eq 3, reduant ")".*
> >
>
> Thanks for your advice. We have fixed that in our manuscript.
>
> > Weakness. 3. *Some usage of notation is not standard. For example, the function "B" accepts three kinds of input.*
> >
>
> Thanks for your advice. We have fixed that in our manuscript.
>
> > Weakness. 4. In line 231, the notation $C_i$ should be $c_i$.
> >
>
> Thanks for your advice. We have fixed that in our manuscript.
>
> > Weakness. 5. *The number of K when selecting harmful samples, and the bias threshold seems not explained in the experiments.*
> >
>
> Thanks for your comments.  In our experiment, we select the number of samples **k=5000** for Colored MNIST, and **k=200** for both Adult and CelebA.
>
> We present experiments of debiasing using Eq. 9 with different values of K (on Colored MNIST with a bias ratio of 0.99) as an example. It can be observed that accuracy and bias improve gradually and tend to be saturated as the number of samples increases, while the time for unlearning (debiasing) continually increases. Hence, in experiments, **we select the number of samples based on both the performance and time cost.**
>
> | Method  | # of samples (K) | Acc.(%) ↑ | Time(s) | Bias ↓ |
> | --- | --- | --- | --- | --- |
> | Vanilla | N/A | 51.34 | - | 0.4931 |
> | Ours | 500 | 73.67 | 3.750s | 0.2757 |
> |  | 1000 | 75.70 | 7.396s | 0.2648 |
> |  | 2000 | 77.36 | 14.60s | 0.2451 |
> |  | 3000 | 78.98 | 25.37s | 0.2232 |
> |  | 5000 | 80.04 | 48.37s | 0.2042 |
> |  | 6000 | 80.06 | 58.24s | 0.2044 |
> |  | 8000 | 80.31 | 84.90s | 0.2044 |
> |  | 10000 | 80.42 | 105.25s | 0.2035 |
>
> **The bias threshold** varies across tasks and could be selected according to practical requirements. In our experiment, we perform the debiasing phase on all tasks to showcase our proposed method (**assuming a threshold of 0**). However, in practical applications, the debiasing phase may not always be necessary. For instance, if we set the threshold to 0.3, in our experiment on Colored MNIST with a bias ratio of 0.95, the bias of the vanilla model (0.2589) is already compliant.

---

> > ### Comment · Reviewer_NRrd · 2023-08-15
> >
> > Thanks for your response, the rebuttal addresses my concerns. I would like to increase my score.

---

> > > ### Author Response · Authors · 2023-08-16
> > >
> > > Dear Reviewer NRrd,
> > >
> > > We are very delighted with your recognition of our paper and rebuttal! Thanks very much for your time and comments! We will fix the bugs and clarify the parameters in our revised manuscript.
> > >
> > > Thank you very much! Best wishes to you!
> > >
> > > Warm regards,
> > >
> > > The Authors

---

### Official Review · Reviewer_2Mrc · 2023-07-06

**Soundness:** 3 good
**Presentation:** 3 good
**Contribution:** 3 good
**Rating:** 8
**Confidence:** 4

**Summary:**

This paper presents a novel method for model debiasing, based on the techniques of counterfactual biases and machine unlearning. The method designs an all-inclusive debiasing pipeline: bias identification, biased-effect evaluation, and bias removal. The proposed method can get rid of time-consuming human labeling or re-training, which is appealing to practical applications. Experimental results and ablation analysis suggest that the proposed method is effective and more efficient on multiple datasets for fairness evaluation. Overall, I appreciate the idea of using machine unlearning and counterfactual samples for efficient model debiasing, though some concerns still need to be addressed for improved quality.

**Strengths:**

Strengths:
1. Model debiasing and fairness are critical for the deployment of large AI models in practice. The proposed method can identify and remove the bias in a cost-effective manner, which is more suitable for practical applications than previous retraining methods.
2. The idea is novel and interesting. Employing counterfactual samples to identify biases seems quite helpful. Leveraging the machine unlearning techniques with counterfactually constructed datasets for bias removal shows a new promising direction for efficient model debiasing.
3. The sanity check is clear, experiments in the main paper and appendix are quite comprehensive, and the performance in terms of accuracy, fairness and efficiency are promising.

**Weaknesses:**

1. The post-processing baselines are included on the Colored MNIST. I suggest including them for other datasets, i.e., CelebA and Adult.
2. More discussions on how the proposed method can be generalized for large-scale networks or tasks. Existing fairness studies are mainly conducted on small datasets like C-MNIST and networks such as MLP/ResNet. It would be great if the authors can investigate how the proposed unlearning method can be extended to large-scale datasets or models, for examples, is it possible to extend this idea to  large-scale language models. Even some preliminary results or discussions would be highly appreciated.
3. Besides quantitative results in Table 1-3, more straightforward visualizations of experimental results on how the proposed method can give rise to better fairness (less bias) are highly encouraged  to better verify the effectiveness.

**Questions:**

See weakness.
Minor issues:
1. Writing need to be improved, e.g., typos and grammar errors: line 40 involves (involve), line 35 “or”, etc.
2. Paper title and manuscript title are not the same. Title should be "Fast model debiasing with machine unlearning"?
3. What are the definitions of  Demographic parity bias (De.) [34], and Equal opportunity bias, and how they relate to the counterfactual bias? The Eq 10 and 11 need to be further clarified.

**Limitations:**

Authors addressed the limitations.

---

> ### Author Rebuttal · Authors · 2023-08-09
>
> >Weakness. 1. *post-processing*
>
> Thanks for your advice. We added the performance of post-processing on CelebA for reference. However, these post-processing methods aim to achieve group fairness by simply altering the predictions of a few individuals and cannot achieve comparable performance.
>
> | Target Attr | Bias Attr | Method | Avg. Acc | Bias |
> | --- | --- | --- | --- | --- |
> | Blonde hair | male | Vanilla | 94.90 | 0.4211 |
> |  |  | DRO | 92.90 | 0.3206 |
> |  |  | LfF | 93.52 | 0.2557 |
> |  |  | LDR | 86.67 | 0.3126 |
> |  |  | EqOdd | 91.03 | 0.4095 |
> |  |  | CEqOdd | 91.84 | 0.4271 |
> |  |  | Reject | 90.87 | 0.3290 |
> |  |  | Ours | 93.41 | 0.0717 |
> | Attractive | male | Vanilla | 77.42 | 0.3695 |
> |  |  | DRO | 78.35 | 0.3004 |
> |  |  | LfF | 77.24 | 0.2815 |
> |  |  | LDR | 81.70 | 0.2986 |
> |  |  | EqOdd | 77.03 | 0.3689 |
> |  |  | CEqOdd | 76.79 | 0.3718 |
> |  |  | Reject | 76.09 | 0.3471 |
> |  |  | Ours | 80.99 | 0.1273 |
>
> >Weakness. 2. *scale to large model*
>
> Thanks for your comments. We further extended our method to LLM (large language model) debiasing scenario. We make use of pre-trained BERT [1] and GPT-2 [2], provided by Huggingface [3]. We use StereoSet [4] as our test set. StereoSet is a large-scale natural dataset to measure stereotypical biases in gender, profession, race, and religion. It reports two metrics: **Language Modeling Score (LMS)** measures the percentage of instances in which a language model prefers the meaningful over meaningless association. **The LMS of an ideal language model will be 100 (the higher the better)**. **Stereotype Score (SS)** measures the percentage of examples in which a model prefers a stereotypical association over an anti-stereotypical association. **The SS of an ideal language model will be 50 (the closer to 50 the better)**.
>
> We use Crows-Pairs [5] as our external dataset. Each sample in Crows-Pairs consists of two sentences: one that is more stereotyping and another that is less stereotyping, which can be utilized as counterfactual pairs. Following the setting in the main paper, we freeze all but the top layer in the trained large model. Results are presented in the following table. We compare our method with five commonly used LLM debiasing baselines: Counterfactual Data Augmentation (CDA) [6], Dropout [7], Iterative Null-space Projection (INLP) [8], Self-debias [9], and SentenceDebias [10].
>
> Results on BERT are presented in the following table (**Results on GPT-2 are presented in Tab.3 in the PDF due to the space constraints**), and show that our method can outperform or achieve comparable performance with baseline methods. **As for BERT, our method reaches best (denoted by “bold”) or second best (denoted by “underline”) performance in 5 of 6 metrics.**
>
> | backbone | attribute | method |SS|LMS|
> | --- | --- | --- | --- | --- |
> | BERT | gender | Vanilla|60.28|84.17|
> |  |  | CDA | 59.61 | 83.08 |
> |  |  | Dropout  | 60.66 | 83.04 |
> |  |  | INLP | **57.25** | 80.63 |
> |  |  | Self-debias | 59.34 | 84.09 |
> |  |  | SentenceDebias | 59.37 | 84.20 |
> |  |  | Ours | $\underline{57.77}$ | **85.45** |
> | BERT | race | Vanilla| 57.03 | 84.17 |
> |  |  | CDA | 56.73 | 83.41 |
> |  |  | Dropout| 57.07 | 83.04 |
> |  |  | INLP | 57.29 | 83.12 |
> |  |  | Self-debias | **54.30** | **84.24**|
> |  |  | SentenceDebias | 57.78 | 83.95 |
> |  |  | Ours | 57.24 | $\underline{84.19}$ |
> | BERT | religion | Vanilla| 59.70 | 84.17 |
> |  |  | CDA | 58.37 | 83.24 |
> |  |  | Dropout  | 59.13 | 83.04 |
> |  |  | INLP | 60.31 | 83.36 |
> |  |  | Self-debias | **57.26** | 84.23 |
> |  |  | SentenceDebias | 58.73 | 84.26 |
> |  |  | Ours | $\underline{57.85}$ | **84.90** |
>
>
>
> >Weakness. 3. *visualization*
>
> Thanks for your advice. We have included presentative images in Fig. 1 in the PDF.
>
>
> >Q1. *typo*
>
> Thanks for your advice. We have fixed that in our manuscript.
>
> >Q2. *title*
>
> We are really sorry for the mistake. The title should be "Fast model debiasing with machine unlearning". We will modify the title.
>
> >Q3. *related works*
>
> Thank for your comments. We will incorporate a more detailed explanation into the related works section. The definitions are as follows:
>
> Demographic Parity [11]:  A predictor $Y$ satisfies demographic parity  if
> $P(Y |A = 0) = P(Y|A = 1)$, where $A$ is the sensitive attribute. The likelihood of a positive outcome
> should be the same regardless of whether the person is in the protected (e.g., female) group.
>
> Equal Opportunity [12]:  “A binary predictor $Y$ satisfies equal opportunity with respect
> to $A$ and $Y$ if $P(Y=1|A=0,Y=1) = P( Y=1|A=1,Y=1)$” . This means that the probability of a person in a positive class being assigned to a positive outcome should be equal for both protected and unprotected (female and male) group members.
>
> The counterfactual fairness definition is based on the intuition that “a
> decision is fair towards an individual if it is the same in both the actual world and a counterfactual world where the individual belonged to a different **demographic group**.” It can be viewed as individual-level Demographic Parity [13].
>
> [1] Bert: Pre-training of deep bidirectional transformers for language understanding.
>
> [2] Language models are unsupervised multitask learners.
>
> [3] Huggingface's transformers: State-of-the-art natural language processing.
>
> [4] StereoSet: Measuring stereotypical bias in pretrained language models.
>
> [5] CrowS-pairs: A challenge dataset for measuring social biases in masked language models.
>
> [6] Counterfactual data augmentation for mitigating gender stereotypes in languages with rich morphology.
>
> [7] Measuring and reducing gendered correlations in pre-trained models.
>
> [8] Null it out: Guarding protected attributes by iterative nullspace projection.
>
> [9] Self-diagnosis and self-debiasing: A proposal for reducing corpus-based bias in nlp.
>
> [10] Towards debiasing sentence representations.
>
> [11] Fairness through awareness.
>
> [12] Equality of opportunity in supervised learning.
>
> [13] Counterfactual Fairness Is Basically Demographic Parity.

---

> > ### Comment · Reviewer_2Mrc · 2023-08-15
> > **Comments**
> >
> > Thanks for the response. The authors provided satisfactory experimental results regarding the large-scale models (Pretrained Bert and GPT-2), and the performance looks quite promising. They also include additional comparisons with baseline and clarifications of related works.
> >
> > I have checked other reviewer's comments and the author's responses. There are several common questions, e.g., adding more baselines, extension to large-scale models, computational complexity etc. The authors have provided additional experimental results and discussions. I agree with other reviewers that the paper could be better enhanced if these additional experimental results in the rebuttal  could be incorporated into the revised manuscript, for example, how to extend to large-scale neural networks, how to leverage existing bias auditing/distribution-shifting datasets (e.g., BERT and GPT-2 with Crows-Pairs), and further extension to discover the key modules for debiasing.
> >
> > Overall, the rebuttal addressed most of my concerns, while the writing could be further improved. Based on the overall quality of the paper and rebuttal, I'd like to keep (maybe increase) my score.

---

> > > ### Author Response · Authors · 2023-08-15
> > > **Replying to Reviewer 2Mrc**
> > >
> > > Dear Reviewer 2Mrc,
> > >
> > > Thank you for your valuable feedback. We are grateful for your recognition of our response and the experiments in the rebuttal. We also highly appreciate your consideration of the comments given by other reviewers in your overall assessment. We agree with your suggestion that incorporating the additional experiments (extension to large models and bias auditing/distribution-shifting datasets) and discussions (insights from influence functions) into our revised manuscript is crucial. We are committed to refining our writing accordingly. Please accept our apologies for any confusion caused.
> > >
> > > Once again, we extend our sincere gratitude for your time and insights. Best wishes to you!
> > >
> > > Warm regards,
> > >
> > > The Authors

---

### Official Review · Reviewer_AWfs · 2023-07-07

**Soundness:** 2 fair
**Presentation:** 3 good
**Contribution:** 3 good
**Rating:** 6
**Confidence:** 5

**Summary:**

This paper presents a novel approach to addressing bias in models by utilizing a small external dataset for de-biasing. The proposed method effectively identifies and mitigates inherent biases by analyzing the disparities between original images and counterfactual images. Initially, the method determines the specific type of bias present in the model by computing these differences. Subsequently, it identifies samples with the highest influence scores in terms of counterfactual bias when a particular bias type is detected. To mitigate this bias, the method updates the model by unlearning these influential samples, using the difference between the gradients of the original and counterfactual samples. The method is evaluated using one synthetic dataset and two real-world datasets. The results demonstrate that the proposed method achieves state-of-the-art performance in mitigating counterfactual bias, while also maintaining low computational costs.

**Strengths:**

* The intuition of the proposed method seems clear and intuitive. Also, the proposed method is novel and well-designed two recent techniques in other domains, influence function and unlearning.

* The proposed method can mitigate the bias of a model efficiently. It only needs a small amount of group-labeled data points. Furthermore, the method requires much less training time than the existing de-biasing methods based on re-training.

* The method achieves good performance on CelebA and Colored MNIST.

**Weaknesses:**

* Lack of some baselines. Given that the objective of this method is to achieve counterfactual fairness or individual fairness, it is important for the authors to compare their approach with existing methods [1,2] specifically designed for individual fairness. This would ensure a fair and comprehensive evaluation of their method's performance in terms of counterfactual bias. It should be noted that Group DRO and LfF, the authors adopted as baselines, were originally designed for group robustness, rather than individual-level fairness.

* Challenges in obtaining counterfactual samples. One drawback of the proposed method is that it requires counterfactual samples for each instance in the external dataset. However, acquiring such counterfactual samples for image datasets is not a trivial task. For instance, assuming that images are annotated with 40 attributes, as done for the CelebA dataset, may not always be feasible.
* There should be a more detailed explanation of why the experiment results turned out the way they were presented. Why does your method that archives the counterfactual bias also mitigate the worst-case group accuracy or any other group fairness metrics?
* Improving the writing clarity.
  - It is suggested to provide clearer distinctions between the problems of model bias and fairness, emphasizing that while there is overlap between the two, they are not synonymous.
  - Additionally, the definition of counterfactual fairness in Equation (1) may be incorrect, compared to the definition introduced in [31].
  - Some typos in equations. In lines 158 and 183, is it z_i instead of z_k?

[1] Yurochkin and Sun. SENSEI: Sensitive set invariance for enforcing individual fairness, ICLR, 2021.
[2] Yurochkin et al. Training individually fair ML models with sensitive subspace robustness, ICLR, 2020.

**Questions:**

* Why is the worst-case accuracy better than the baselines in Table 2? Why does unlearning by Co. outperform unlearning by De. in terms of De?

* Why does unlearning by De. fail to better demographic parity bias in Table 4?

* In Table 7, unlearning more layers of neural networks does not lead to better performance. Why?

**Limitations:**

The author has addressed their limitations.

---

> ### Author Rebuttal · Authors · 2023-08-09
>
> We thanks the reviewer for the constructive comments. Below are point-to-point responses (W denotes Weakness).
>
> >W1 *baselines*
>
> We did experiment with the two individual fairness baselines.  The baselines can achieve good individual fairness while maintaining stable accuracy. However, they demand the entire annotated dataset for retraining, resulting in a much higher debiasing cost.
>
> |Attribute| Method|Acc(%) ↑|Bias ↓|Time(s)|
> | --- | --- | ---|---|---|
> |gender| Vanilla | 85.40 | 0.0195 |-|
> | | SenSR | 84.09 | 0.0049 | 571 |
> | | SenSeI | 83.91 | 0.0016 | 692 |
> | | Ours | 81.89 | 0.0005 | 2.49 |
> |race|Vanilla| 84.57 | 0.0089 |-|
> |  |SenSR| 84.09 | 0.0036 | 571 |
> |  |SenSeI| 83.91 | 0.0015 | 692 |
> |  |Ours|83.80| 0.0013 | 2.54 |
>
> >W2 *dataset generation*
>
> We agree that most counterfactual fairness auditing work require generating pairs with labeled sensitive attributes [1, 2, 3], while many datasets do not have enough labels.
>
> One possible solution is to directly use other fairness-auditing datasets, instead of building our own one if labels are unavailable. Following your advice, we did experiments with **large language models (LLMs)**, where we use the Crows-Pairs [4] , which measures the social bias in LMs against protected attributes, as “external” datasets. **Results show that our FMD can outperform or achieve comparable performance with baselines on large models (see Response to Reviewer 2Mrc W2).**
>
> In addition, we compute the time cost to construct external datasets. For **Colored MNIST**, it takes **4.3s** to generate the external dataset (5000 samples) on an AMD CPU. Similarly, it takes **0.51s (200 samples)** for
>  **Adult** and **11.2s (200 samples)** for **CelebA**. Counting such cost in, the time cost of baselines is still more than 15 times of ours.
>
> Research on fairness with few/no attribute labels is still in the infant stage [5], and we will further explore it.
>
> >W3 *Worst-group accuracy*
>
> Worst-group accuracy reports the lowest accuracy among all groups. We computed the group statistics and accuracy on CelebA in the table below as an example. Supervised training achieves **94.90%** accuracy on the test set. However, for the **{blonde, male}** group, the accuracy is only **47.36%**. This indicates that the model captures the unintended decision rule between hair color and gender.
>
> Our method utilizes counterfactual samples to unlearn the unintended decision rule that the model has learned. The **38% improvement** in worst-group accuracy demonstrates its effectiveness. Moreover, the **smaller accuracy disparity** among different groups indicates that the predictions are more fair.
>
> The reviewer's question may also revolve around why individual fairness metrics (Counterfactual) could lead to better group fairness (worst-group accuracy). Basically, our method reduces the mispredictions for each **{blonde, male}** individual, thereby leading to better group accuracy. **Additionally, as pointed out in [1], the individual and group fairness measures donot necessarily reflect different normative principles.**
>
> | |# of samples| |Acc (%)| |
> |---|---|---|---|---|
> |Group|Train|Test|Vanilla|Ours|
> |Blond=0, Male=0|71629|9767|99.32|97.80|
> |Blond=0, Male=1|66874|7535|89.77|89.98|
> |Blond=1, Male=0|22880|2480|94.54|94.01|
> |Blond=1, Male=1|1387|180|47.36|**87.15**|
> |Total|162770|19962|94.90|93.41|
>
> >W4 *clarity*
>
> a) Fairness is a high-level category, indicating the absence of any prejudice or favoritism toward an individual or group [7]. Here we focus on the bias (unfairness) learned by ML models. We will clarify the definitions.
>
> b) Sorry for the confusion. It should be $P(\hat{Y}_{A←a} = y | X = x, A = a) = P(\hat{Y}\_{A←\bar{a}} =y | X = x, A = a)$, where $y = f\_{\hat{\theta}}(X,A)$, to imply the process of attribute changing. Note that we assume that $X$ are non-descendants of $A$ so that $X\_{A←a} = X\_{A←\bar{a}}$.
>
> c) We fixed the typos.
>
> >Q1
>
>  See W3.
>
> >Q2 *De. & Co.*
>
> Different from counterfactual bias, demographic parity bias measures the difference in predictions (of a model) on different protected demographic **groups**, e.g., gender or race. One reason is that when scaling to a group, influence estimation may not be accurate, so that the selected harmful samples may not be effective. To validate it, we select top 1500 samples and count the number of bias-aligned samples (harmful). The number of De. is **1153** and of Co. is **1465**, which indicating Co. is **more effective on selecting harmful samples**.
>
> >Q3 *multi-layer estimation*
>
> Previous work investigates the estimation accuracy of the influence function on both multi-layer and single-layer setups [8]. It performs a case study on the MNIST.  For each test point, they select 100 training samples and compute the ground-truth influence by model re-training. **Results show that estimations are more accurate for shallow networks.**
>
> Our results in Tab 7 also validate this point. When applying FMD to a three-layer neural net, the performance on either accuracy or bias becomes worse. This could potentially be attributed to the inaccurate estimation of influence function on multi-layer neural nets. In our experiments, we adhere to the set-up in [9], where the influence function is only applied to the last layer of deep models, which proves to be effective.
>
> [1] Explaining machine learning classifiers through diverse counterfactual explanations
>
> [2] Auditing fairness under unawareness through counterfactual reasoning
>
> [3] Counterfactual explanations for robustness, transparency, interpretability, and fairness of artificial intelligence models
>
> [4] CrowS-pairs: A challenge dataset for measuring social biases
>
> [5] Unsupervised learning of debiased representations with pseudo-attributes
>
> [6] On the apparent conflict between individual and group fairness
>
> [7] A survey on bias and fairness in machine learning
>
> [8] Influence functions in deep learning are fragile
>
> [9] Understanding black-box predictions via influence functions

---

> > ### Author Response · Authors · 2023-08-17
> >
> > Dear Reviewer AWfs,
> >
> > We greatly appreciate your valuable time and thoughtful feedback. We have thoroughly revised our paper and responded in accordance with your suggestions, aiming to address all your concerns. We would be grateful if you would kindly let us know of any other concerns and if we could further assist in clarifying any other issues.
> >
> > Thank you once again for your contributions, and we extend our best wishes.
> >
> > Sincerely,
> >
> > Authors

---

> > ### Comment · Reviewer_AWfs · 2023-08-19
> > **Reply to author's rebuttal**
> >
> > I thank you for your comprehensive rebuttal and apologize for the slight delay in my response. I truly appreciate the novel contribution regarding the proposition of a new post-processing method based on unlearning techniques. However, I still have a few remaining concerns about the current version of the paper that I believe could enhance its overall quality.
> >
> > 1. I understand from your response that individual and group fairness can sometimes align, yet they may also present conflicts, as mentioned in [A1].  To ensure a thorough and fair comparison, I recommend explicitly delineating the baseline methods in accordance with the fairness notion they target. Given that your method considers individual fairness, I suggest focusing the initial comparison on those baseline methods explicitly designed for individual fairness, such as SenSEI or SenSR, since the baseline methods used in the original paper, such as LDR, LfF, Rebias, and DRO, implicitly or explicitly target group fairness. Furthermore, considering post-processing methods like [A2] that target individual fairness could contribute to a more comprehensive analysis.
> >
> > 2. While the authors have included supplementary results in NLP domains, the applicability of the method remains somewhat constrained to diverse vision applications. because it is extremely hard to generate counterfactual images.  In the paper, the considerations around Colored MNIST and CelebA datasets, do not fully represent real-world counterfactual scenarios. Colored MNIST considers just a synthetic bias we can control. Also, image pairs in CelebA are not real counterfactual images (in Figure 1(b), the backgrounds in the image pairs are different). However, the proposed method heavily relies on counterfactual samples in order to detect the bias and compute the influence score. Hence, it would greatly enhance the paper's rigor to address this limitation more extensively.
> >
> > 3. Building on the previous point, the use of potentially incomplete counterfactual samples in CelebA brings about an associated limitation in the bias metric's completeness. I believe it would be beneficial for the paper to delve deeper into this matter, outlining the potential implications and strategies for addressing such incompleteness.
> >
> > 4. If the estimation of the influence score is not accurate in deeper networks as shown in [8], how do we believe that the influence score for the last layer in DNN models is accurate? My concern regarding this point has been also highlighted in [A3], which shows that influence score estimation may be a poor match to leave-one-out retraining for nonlinear networks. As this concern could impact your method's performance, I encourage a more thorough discussion on how your approach mitigates or accounts for this challenge.
> >
> > 5. One important baseline [A4] is missing. [A4] also proposed a method that identifies negative fairness samples by estimating the influence scores and re-trains a model by re-weighting the training samples based on the scores. The difference from [A4] should be addressed in the paper and used as a baseline method. This inclusion could help provide a more comprehensive view of your method's advantages.
> >
> > Once again, I truly appreciate your efforts and the thought-provoking responses you've shared.
> >
> > [A1]. Dwork et at. Fairness Through Awareness, 2012.
> >
> > [A2]. Petersen et al. Post-processing for Individual Fairness, NeurIPS, 2022.
> >
> > [A3]. Bae et al. If influence Functions are the Answer, Then What is the Question?, NeurIPS, 2022.
> >
> > [A4]. Li and Liu. Achieving Fairness at No Utility Cost via Data Reweighting with Influence, ICML, 2022.

---

> > > ### Author Response · Authors · 2023-08-20
> > > **Reply to Reviewer AWfs (1/3)**
> > >
> > > We greatly value your expert feedback on our paper. We hope that our responses can address any concerns you may have.
> > >
> > > > 1. I understand from your response that individual and group fairness can sometimes align, yet they may also present conflicts, as mentioned in [A1]. To ensure a thorough and fair comparison, I recommend explicitly delineating the baseline methods in accordance with the fairness notion they target. Given that your method considers individual fairness, I suggest focusing the initial comparison on those baseline methods explicitly designed for individual fairness, such as SenSEI or SenSR, since the baseline methods used in the original paper, such as LDR, LfF, Rebias, and DRO, implicitly or explicitly target group fairness. Furthermore, considering post-processing methods like [A2] that target individual fairness could contribute to a more comprehensive analysis.
> > >
> > > We compare our method with one pre-processing baseline [A4], 6 in-processing debiasing baselines (LDR, LfF, Rebias, DRO, SenSEI and SenSR) and 4 post-processing baselines (EqOdd [B1], CEqOdd [B2], Reject [B3] and [A2]). [A4] utilizes influence function to reweight the training sample, in order to re-train a fair model targeting group fairness metrics (equal opportunity and demographic parity). Among in-processing baselines, LDR, LfF, Rebias, and DRO are designed explicitly targeting higher accuracy (on unbiased test set or worst-group test set) and implicitly targeting fairness, while SenSEI and SenSR are designed targeting individual fairness. EqOdd, CEqOdd and Reject are designed targeting different group fairness metrics (equal odd and demographic parity), while [A2] propose a post-processing algorithms for individual fairness.
> > >
> > > For fair comparisons, we include performance on both accuracy and fairness metrics. In our paper, we use counterfactual fairness, which provides explicit causal explanations for individual fairness. Furthermore, it's worth noting that, as discussed in [B4], counterfactual fairness also reflects group fairness.
> > >
> > > Results of additional baselines on Adult are provided below. We will further incorporate new baselines on other datasets.
> > >
> > >
> > > | Attribute | Method | Category | Fairness Target | Acc.(%) ↑ | Bias ↓ | Time(s) |
> > > | --- | --- | --- | --- | --- | --- | --- |
> > > | gender | Vanilla | - | - | 85.40 | 0.0195 | - |
> > > |  | Reweigh [A4]  | pre-processing | group fairness | 82.60 | 0.0051 | 36 |
> > > |  | SenSR | in-processing | individual fairness | 84.09 | 0.0049 | 571 |
> > > |  | SenSeI | in-processing | individual fairness | 83.91 | 0.0016 | 692 |
> > > |  | EqOdd | post-processing | group fairness | 82.27 | 0.0157 | 0.01 |
> > > |  | CEqOdd | post-processing | group fairness | 82.94 | 0.0039 | 0.6 |
> > > |  | Reject | post-processing | group fairness | 73.37 | 0.0574 | 13 |
> > > |  | PP-IF [A2] | post-processing | individual fairness | 81.96 | 0.0027 | 13 |
> > > |  | Ours | post-processing | individual fairness | 81.89 | 0.0005 | 2.49 |
> > > | race | Vanilla | - | - | 84.57 | 0.0089 | - |
> > > |  | Reweigh [A4] | pre-processing | group fairness | 82.97 | 0.0015 | 36 |
> > > |  | SenSR | in-processing | individual fairness | 84.09 | 0.0036 | 571 |
> > > |  | SenSeI | in-processing | individual fairness | 83.91 | 0.0015 | 692 |
> > > |  | EqOdd | post-processing | group fairness | 83.72 | 0.0149 | 0.01 |
> > > |  | CEqOdd | post-processing | group fairness | 83.06 | 0.0023 | 3.6 |
> > > |  | Reject | post-processing | group fairness | 75.19 | 0.1038 | 14 |
> > > |  | PP-IF [A2] | post-processing | individual fairness | 82.37 | 0.0015 | 13 |
> > > |  | Ours | post-processing | individual fairness | 83.80 | 0.0013 | 2.54 |
> > >
> > >
> > >
> > > [B1] Hardt, Moritz, Eric Price, and Nati Srebro. "Equality of opportunity in supervised learning." *Advances in neural information processing systems* 29 (2016).
> > >
> > > [B2] Pleiss, Geoff, et al. "On fairness and calibration." *Advances in neural information processing systems* 30 (2017).
> > >
> > > [B3] Kamiran, Faisal, Asim Karim, and Xiangliang Zhang. "Decision theory for discrimination-aware classification." *2012 IEEE 12th international conference on data mining*. IEEE, 2012.
> > >
> > > [B4] Rosenblatt, Lucas, and R. Teal Witter. "Counterfactual Fairness Is Basically Demographic Parity." *Proceedings of the AAAI Conference on Artificial Intelligence*. Vol. 37. No. 12. 2023.

---

> > > > ### Author Response · Authors · 2023-08-20
> > > > **Reply to Reviewer AWfs (2/3)**
> > > >
> > > > > 2. While the authors have included supplementary results in NLP domains, the applicability of the method remains somewhat constrained to diverse **vision applications**. because it is extremely hard to generate counterfactual images. In the paper, the considerations around Colored MNIST and CelebA datasets, do not fully represent real-world counterfactual scenarios. Colored MNIST considers just a synthetic bias we can control. Also, image pairs in CelebA are not real counterfactual images (in Figure 1(b), the backgrounds in the image pairs are different). However, the proposed method heavily relies on counterfactual samples in order to detect the bias and compute the influence score. Hence, it would greatly enhance the paper's rigor to address this limitation more extensively.
> > > >
> > > > In our experiments, we utilize approximated counterfactual samples for CelebA due to the unavailability of strict counterfactual data. Based on attribute annotations, we select images with the same target attributes but opposite sensitive attributes, while maintaining other attributes as much as possible. Our method achieves the best results on the worst-case group, indicating that the approximated counterfactual samples can also effectively enhance fairness in predictions. Similar to our approach, [B5] proposes to select pairs of counterfactual images based on attribute annotations on the CUB dataset to produce counterfactual visual explanations. Their experiments also show that neural networks can discern major differences (such as gender in our work) between images without strict control (such as background).
> > > >
> > > > For real-world visual datasets (like facial dataset or ImageNet), the unavailability of strict counterfactual data is a common challenge. Existing methods propose to train a generative model to create counterfactual images with altered sensitive attributes [B6, B7, B8, B9], which seems to be a viable approach for obtaining counterfactual datasets for more diverse vision applications. Building upon these methods, we will extend our approach to more scenarios.
> > > >
> > > >
> > > >
> > > > > 3. Building on the previous point, the use of potentially incomplete counterfactual samples in CelebA brings about an associated limitation in the bias metric's completeness. I believe it would be beneficial for the paper to delve deeper into this matter, outlining the potential implications and strategies for addressing such incompleteness.
> > > >
> > > > Thank you for pointing this out. Our measured counterfactual bias on CelebA is not rigorous as the external dataset is not strictly counterfactual. The rigorous definition should be “for individuals in different demographic group (e.g. gender), a fair model should give similar predictions on similar individuals (w.r.t. other attributes)”, which reflects individual fairness requests.
> > > >
> > > >
> > > > [B5] Goyal, Yash, et al. "Counterfactual visual explanations." *International Conference on Machine Learning*. PMLR, 2019.
> > > >
> > > > [B6] Dash, Saloni, Vineeth N. Balasubramanian, and Amit Sharma. "Evaluating and mitigating bias in image classifiers: A causal perspective using counterfactuals." *Proceedings of the IEEE/CVF Winter Conference on Applications of Computer Vision*. 2022.
> > > >
> > > > [B7] Kim, Hyemi, et al. "Counterfactual fairness with disentangled causal effect variational autoencoder." *Proceedings of the AAAI Conference on Artificial Intelligence*. Vol. 35. No. 9. 2021.
> > > >
> > > > [B8] Joo, Jungseock, and Kimmo Kärkkäinen. "Gender slopes: Counterfactual fairness for computer vision models by attribute manipulation." *Proceedings of the 2nd international workshop on fairness, accountability, transparency and ethics in multimedia*. 2020.
> > > >
> > > > [B9] Cheong, Jiaee, Sinan Kalkan, and Hatice Gunes. "Counterfactual fairness for facial expression recognition." *European Conference on Computer Vision*. Cham: Springer Nature Switzerland, 2022.

---

> > > > > ### Author Response · Authors · 2023-08-20
> > > > > **Reply to Reviewer AWfs (3/3)**
> > > > >
> > > > > > 4. If the estimation of the influence score is not accurate in deeper networks as shown in [8], how do we believe that the influence score for the last layer in DNN models is accurate? My concern regarding this point has been also highlighted in [A3], which shows that influence score estimation may be a poor match to leave-one-out retraining for nonlinear networks. As this concern could impact your method's performance, I encourage a more thorough discussion on how your approach mitigates or accounts for this challenge.
> > > > >
> > > > > As verified in [B10], [8] and [A3],  influence estimation matches closely to leave-one-out retraining for **logistic regression model**. As discussed in [B6], measuring influence score for the last layer can be regarded as calculating influence from a logistic regression model on the bottleneck features (Sec. 5.1). The same setup is followed by many influence function-based works [B11, B12] and proves to be effective.
> > > > >
> > > > >
> > > > > > 5. One important baseline [A4] is missing. [A4] also proposed a method that identifies negative fairness samples by estimating the influence scores and re-trains a model by re-weighting the training samples based on the scores. The difference from [A4] should be addressed in the paper and used as a baseline method. This inclusion could help provide a more comprehensive view of your method's advantages.
> > > > >
> > > > > The main difference between our approach and [A4] lies in the fact that [A4] necessitates the computation of two influence functions for each training sample, making it challenging to apply to large models and extensive training sets due to computational costs ([A4] performs experiments on three simple tabular datasets with two-layer neural networks). In contrast, our method circumvents this issue by post-hoc unlearning with an external dataset. Furthermore, [A4] is a pre-processing technique, requiring an additional re-training step after computing the influence for reweighting the training samples.
> > > > >
> > > > > We incorporate [A4] as our baseline, with results provided in the Table. On Adult, it can be observed that [A4] cannot achieve better fairness than our method, even with more procedures.
> > > > >
> > > > > Once again, we extend our heartfelt gratitude for your time and the invaluable suggestions you've provided! We would be grateful if you would inform us of any remaining concerns or questions.
> > > > >
> > > > > [B10] Koh, Pang Wei, and Percy Liang. "Understanding black-box predictions via influence functions." *International conference on machine learning*. PMLR, 2017.
> > > > >
> > > > > [B11] Pruthi, Garima, et al. "Estimating training data influence by tracing gradient descent." *Advances in Neural Information Processing Systems* 33 (2020): 19920-19930.
> > > > >
> > > > > [B12] Yeh, Chih-Kuan, et al. "Representer point selection for explaining deep neural networks." *Advances in neural information processing systems* 31 (2018).

---

> > > > > > ### Author Response · Authors · 2023-08-21
> > > > > > **Results of additional experiments have been updated**
> > > > > >
> > > > > > Dear Reviewer AWfs,
> > > > > >
> > > > > > Results of additional experiments have been updated in Reply to Reviewer AWfs (1/3). We hope that our responses can address your concerns. We would be grateful if you would inform us of any remaining concerns or questions.
> > > > > >
> > > > > > Thank you once again for your comments, and we extend our best wishes.
> > > > > >
> > > > > > Sincerely,
> > > > > >
> > > > > > Authors

---

> > > > > > > ### Comment · Reviewer_AWfs · 2023-08-21
> > > > > > > **Reply to Author's Response**
> > > > > > >
> > > > > > > I sincerely appreciate your patient response! While I am still a little skeptical about the assumption that counterfactual images can be collected or generated, your other responses mostly solved my concerns and helped improve my understanding of the authors' method. I will increase my score and agree that this paper is enough to be accepted. I wish the additional results and discussion will be included in the camera-ready version.

---

> > > > > > > > ### Author Response · Authors · 2023-08-21
> > > > > > > >
> > > > > > > > Thank you very much for your comprehensive comments and suggestions! We sincerely appreciate them and we will make revisions to our manuscript based on your comments and discussions in order to further improve the quality of our work.
> > > > > > > >
> > > > > > > > Thanks again and best wishes!

---

### Author Rebuttal · Authors · 2023-08-09

Dear reviewers,

We greatly appreciate your valuable comments and the time and dedication you invested in reading, comprehending, and assessing our paper. We thoroughly reviewed all the questions, weaknesses raised by the reviewers and addressed them accordingly. Several questions have attracted the attention of multiple reviewers, such as **scale to large model** (R. 2Mrc W2, hEy4 and hkdb) , **number of K** (R. NRrd W5, hEy4 and hkdb) and **dataset generation** (R. AWfs W2&hkdb). **Due to space limits, we only placed the response to each first corresponding reviewer. We apologize for the inconvenience caused to other reviewers.** We have provided comprehensive responses to each review individually, and we are confident that our responses adequately address all the concerns raised by the reviewers. We are in the process of updating and uploading both the paper and supplementary materials to incorporate our responses. Should further details, explanations, or clarifications be required, we are more than willing to provide them.

Thank you once again for your thoughtful feedback.

**Due to the space limitation, we continue the response to R. hkdb below. Sorry for the bothering to other reviewers.**

**Response to R. hkdb:**

> WA-f. *influence function*:

We will incorporate this explanation into the article. Our Bias-effect evaluation is based on the influence functions [1]:
$I\_{up, params}(z)  = \frac{d\hat\theta\_{\epsilon, z}}{d\epsilon}\Bigr|\_{\substack{\epsilon = 0}}.$
It gives an efficient approximation to compute the parameter change if $z$ were upweighted (by some small $\epsilon$).

In our paper, our goal is to quantify how removing each training point $z$ contributes to bias. We apply the chain rule:
$I_{up, bias}(z, B(\hat{\theta})) = \frac{dB(\hat\theta_{\epsilon, z})}{d\hat\theta_{\epsilon, z}} \frac{d\hat\theta_{\epsilon, z}}{d\epsilon}\Bigr|_{\substack{\epsilon = 0}}.$
Intuitively, this equation can be understood in two parts: the latter part calculates the impact of removing $z$ on the parameters. The former part corresponds to the derivative of bias with respect to parameters, assessing how changes in parameters affect the bias. Hence, this equation quantifies the influence of removing $z$ on the bias.

> WA-g. *baselines*:

We compare our method with four in-processing debiasing baselines. We further include two individual-fairness baselines and three post-processing debiasing baselines following the suggestion of R. AWfs and 2Mrc. LDR [2] proposes a feature augmentation approach via disentangled representation for debiasing. Rebias [3] proposes to train a de-biased representation by encouraging it to be different from a set of biased representations. LfF [4] proposes to firstly train a biased network, then debias the second network by focusing on samples that go against the biased network. Distributionally robust optimization (DRO) [5] proposes to improve the worst-case generalization performance to perform unbiased optimization. We will include these details in the supplementary section.

> WB. *terminology*:

This is an insightful perspective. In our proposed method (Eq. (9)), what we actually eliminate is the influence brought by the bias attribute, rather than the impact of the sample itself. Therefore, "debiasing" would be more appropriate. We will replace "unlearn" with "debias" in our method description.

> WC-a. *confidence intervals*:

We have added confidence intervals in Tab.1 in the PDF.

> WC-b. *DRO & Unb*:

Group Distributionally robust optimization (DRO) [5] has been proposed to improve the worst-case generalization performance. In [5], experiments are all performed on binary classification tasks with binary attributes (i.e. 4 sub-groups).
When extending DRO to Colored MNIST, the number of groups to form is $10*10$, which will not be efficient to optimize. We added the performance of DRO on Colored MNIST (in Tab.1 in the PDF). The results indicate that DRO is not a robust baseline.

We also add the results on Colored MNIST for **Wor.** (testing on bias-conflicting samples) and **Unb.** (testing on 50% bias-aligned and 50% bias-conflicting samples).

> WC-c. *scale to large model*:

See response to R. 2Mrc W2 due to the space limit.

> WC-d. *oracle*:

Thank you for your suggestion. We have included the "Oracle" results in Tab.1 in the PDF. Following your advice, we identified harmful samples using influence functions, excluded an equal number of samples from the training set, and then retrained the model. It can be observed that "Oracle" is better than our method. This outcome might be attributed to errors in influence estimation. However, it's important to note that computing influence on the training set and subsequent retraining is time-consuming (27 times of ours).

> WC-e. *debiasing strategies*:

In Tab.5, we compare Eq. 7 (unlearning training samples), Eq. 8 (unlearning training samples with counterfactual pair), and Eq. 9 (unlearning external counterfactual pairs). Comparing Eq. 7 and Eq. 8, we find that under the same other settings, introducing counterfactual samples to remove bias proves to be a more efficient approach. Furthermore, Eq. 9 serves as an alternative solution to Eq. 8 when training data is unavailable. Empirical evidence indicates that the performance of Eq. 9 is slightly lower than that of Eq. 8.

Directly comparing Eq.7 and Eq.9 could encounter issues due to non-uniform variables. However, intuitively, the observation that Eq.7 performs worse than Eq.9 suggests that debiasing by counterfactual samples is more efficient than directly unlearning harmful training data.


[1] Understanding black-box predictions via influence functions.

[2] Learning debiased representation via disentangled feature augmentation.

[3] Learning de-biased representations with biased representations.

[4] Learning from failure: De-biasing classifier from biased classifier.

[5] Distributionally robust neural networks for group shifts.

---

### Decision · Program_Chairs · 2023-09-21

**Decision:**

Accept (poster)

**Comment:**

This paper presents a novel approach for model debiasing based on machine unlearning and counterfactual fairness. The method can identify and remove the bias with a small external dataset, without requirement for model retraining or time-consuming finetuning. Empirical results show that the proposed FMD can achieve superior debiasing performance across multiple datasets, network architectures and tasks. The paper is well motivated, and provides an extensive set of experiments to demonstrate the efficiency and effectiveness of the proposed method. Thus, this work is of value to the NeurIPS community, and all reviewers agree to accept this paper.